# ONE-EMBEDDING-FITS-ALL: EFFICIENT ZERO-SHOT TIME SERIES FORECASTING BY A MODEL ZOO

## ABSTRACT

The proliferation of Time Series Foundation Models (TSFMs) has significantly advanced zero-shot forecasting, enabling predictions for unseen time series without task-specific fine-tuning. Extensive research has confirmed that no single TSFM excels universally, as different models exhibit preferences for distinct temporal patterns. This diversity suggests an opportunity: how to take advantage of the complementary abilities of TSFMs. To this end, we propose ZooCast, which characterizes each model's distinct forecasting strengths. ZooCast can intelligently assemble current TSFMs into a model zoo that dynamically selects optimal models for different forecasting tasks. Our key innovation lies in the *One-Embedding-Fits-All* paradigm that constructs a unified representation space where each model in the zoo is represented by a single embedding, enabling efficient similarity matching for all tasks. Experiments demonstrate ZooCast's strong performance on the GIFT-Eval zero-shot forecasting benchmark while maintaining the efficiency of a single TSFM. In real-world scenarios with sequential model releases, the framework seamlessly adds new models for progressive accuracy gains with negligible overhead.

## 1 INTRODUCTION

Time-series forecasting is crucial across domains, including finance (Sezer et al., 2020), meteorology (Karevan & Suykens, 2020), industrial systems (Choi et al., 2021; Jin et al., 2024a; Styles et al., 2022), healthcare (Penfold & Zhang, 2013), and environmental science (Karevan & Suykens, 2020), enabling informed decision-making through historical pattern analysis. Recently, the emergence of Time Series Foundation Models (TSFMs) (Ma et al., 2024) has significantly advanced zero-shot forecasting, where a model predicts future values directly from a historical input window without task-specific fine-tuning. However, these independently developed TSFMs vary greatly in architecture and training methods, leading to diverse predictive strengths and weaknesses (Aksu et al., 2024; Dooley et al., 2023; Gruver et al., 2023). For instance, Chronos demonstrates particular strength in high-frequency electricity data (Aksu et al., 2024), while VisionTS significantly outperforms peers on cloud data full of spikes (Toner et al., 2025). Since no single model achieves optimality universally, this naturally leads to an opportunity where the complementary strengths of different models can be combined. To this end, we introduce a model zoo, a curated collection of diverse TSFMs that leverages their complementary abilities. Such a zoo not only enhances forecasting performance beyond any single TSFM but also preserves the efficiency of zero-shot inference, providing a principled direction for building stronger forecasting systems.

How to utilize existing TSFMs in a model zoo for optimal zero-shot performance on unseen forecasting tasks? A simple solution would be enumerating all TSFMs for new tasks or naively ensembling all of them. While these approaches may achieve relatively good performance, contemporary TSFMs typically range from hundreds of megabytes to several gigabytes in size (Chen et al., 2025; Das et al., 2024; Goswami et al., 2024; Woo et al., 2024), making such exhaustive methods computationally prohibitive. We identify the key challenges for model-zoo-based forecasting: how to select among current models to enhance zero-shot performance on arbitrary forecasting tasks while maintaining efficiency, keeping computational overhead acceptable.

To address these challenges, we introduce ZooCast, a novel model-zoo-based paradigm for zero-shot model selection. The *core insight* of ZooCast is to embed both forecasting tasks and TSFMs into a unified low-dimensional vector space, transforming model-task matching into a rapid similarity

Figure 1: Workflow of zero-shot forecasting with model zoo. **Stage 1 (Left)**: A precomputation stage, each TSFM's predictive preferences are summarized into compact embeddings via a co-embedding extractor, stored in a unified zoo repr. library (repr. for representation). **Stage 2 (Right)**: An instant selection stage, new forecasting tasks are rapidly embedded and matched against the precomputed representations through lightweight similarity computations, enabling immediate optimal model recommendation and predictions. This "precompute-and-select" design ensures ZooCast enhances forecasting accuracy while maintaining efficiency, and exhibits scalability for future models.

search. As shown in Figure 1, ZooCast first precomputes each model's predictive preferences exactly once, storing them in a model zoo representation library. When new tasks arrive, their representations are computed and matched against precomputed model representations for instant selection.

A central challenge of model-zoo forecasting is under three difficult conditions. First, characterizing each model's predictive strengths is nontrivial, since different TSFMs excel on distinct temporal patterns and cannot be distinguished without exhaustive evaluation. Second, embedding both tasks and models into a shared space is challenging, as time series exhibit complex non-stationary and multi-channel structures. Third, converting noisy similarity signals into a stable ranking is particularly difficult, as inconsistencies across channels can easily mislead selection. To tackle these challenges, ZooCast introduces three components: (1) advantage subset characterization to capture model-specific strengths with minimal data, (2) model–task co-embedding to place tasks and models in a shared representation space, and (3) error-correcting consensus ranking to ensure robust final selection.

Empirical evaluation demonstrates that ZooCast achieves rapid model selection on GIFT-Eval zero-shot benchmarks. ZooCast-guided predictions outperform individual TSFMs and naive ensembles, achieving superior ranking performance. Surprisingly, we find ZooCast enables additional test-time computation to trade for higher accuracy. For real-world deployment, ZooCast automatically adapts to incrementally released models, continuously improving accuracy as better TSFMs become available. Our contributions are:

- We introduce ZooCast, the first model zoo framework for zero-shot time series forecasting that enhances existing TSFMs' test-time performance.
- A novel model-task co-embedding approach that dramatically reduces selection time while maintaining acceptable computational overhead.
- State-of-the-art performance on zero-shot benchmark with automatic performance gains when integrating newly released models in real-world scenarios.

## 2 RELATED WORK

**Time Series Foundation Models**. Traditional deep learning approaches model temporal dependencies through sequence architectures (Wu et al., 2021; Zhou et al., 2022; Liu et al., 2024; Han et al., 2024), while recent methods adapt LLMs for time series by leveraging heterogeneous pretraining (Garza & Canseco, 2023; Zhou et al., 2023; Jin et al., 2024b; Liang et al., 2024; Cao et al., 2024; Bian et al., 2024). Yet direct transfer remains limited by modality gaps (Tan et al., 2024). Pretrained TSFMs (Dooley et al., 2023; Oreshkin et al., 2021) have shown strong zero-shot forecasting ability with minimal adaptation (Goswami et al., 2024; Das et al., 2024; Chen et al., 2025; Woo et al., 2024; Ansari et al., 2024; Hoo et al., 2024; 2025; Liu et al., 2025), but they differ substantially in architecture and training, while time series tasks vary widely across domains and scales. Thus, even advanced TSFMs cannot ensure universal superiority (Aksu et al., 2024; Dooley et al., 2023; Gruver et al., 2023; Toner et al., 2025), leaving model selection unresolved in real-world deployments.

**Model Selection Methods**. Early transfer learning relied on brute-force fine-tuning, accurate but infeasible at scale. Forward-based methods such as LEEP (Nguyen et al., 2020) and LogME (You et al., 2021) improved efficiency but still required labels and light training, limiting strict zero-shot use. Representation-based approaches, e.g., TASK2VEC (Achille et al., 2019), TS2Vec (Yue et al., 2022), SimMTM (Dong et al., 2023), and TimesURL (Liu & Chen, 2024), map tasks into embeddings but either depend on classification settings, lack transferability guarantees, or remain costly. Learning-to-rank solutions like ModelSpider (Zhang et al., 2023) achieve accuracy with labels but are not extended to regression or forecasting. Overall, current methods suffer from rigidity, task specificity, computational overhead, and missing integrated ranking, leaving time series zero-shot selection an open problem. (See Appendix C for detailed discussion.)

**Model Zoo for Machine Learning**. Above challenge motivates the model zoo solution. Model zoos aggregate diverse models to capture inter-model differences (Yuan et al., 2025), enabling task-specific guidance. They have succeeded in CV and NLP (Zhang et al., 2023; Li et al., 2024; Zhang et al., 2025b), but time series require designs addressing temporal continuity, non-stationarity, multichannel and multifrequency structures. Compared with prior approaches that either relied on costly labeled supervision or lacked mechanisms to capture transferability, recent advances move toward more efficient, forward-free selection strategies that emphasize semantic consistency, robustness to structural heterogeneity. Our work follows this direction and pioneers TSFM-based model zoos for time series, introducing a unified framework that enhances both prediction accuracy and practical usability in low-resource scenarios while maintaining cross-domain flexibility.

## 3 PRELIMINARY

### 3.1 FOUNDATION MODELS FOR MULTIVARIATE TIME-SERIES FORECASTING

Multivariate time-series forecasting aims to predict future values of multiple interdependent variables based on their historical observations. Formally, let $\boldsymbol{X} = [\boldsymbol{x}_1, \ldots, \boldsymbol{x}_T] \in \mathbb{R}^{C \times T}$ denote the historical input with $C$ channels and length $T$, where $\boldsymbol{x}_t \in \mathbb{R}^C$ is the observation at time $t$. Given a forecasting horizon $H$, the task is to predict the future multivariate sequence $\boldsymbol{Y} = [\boldsymbol{x}_{T+1}, \ldots, \boldsymbol{x}_{T+H}] \in \mathbb{R}^{C \times H}$ conditioned on $\boldsymbol{X}$ and $H$, yielding predictions $\hat{\boldsymbol{Y}} = [\hat{\boldsymbol{x}}_{T+1}, \ldots, \hat{\boldsymbol{x}}_{T+H}]$. This formulation makes explicit that the target is the entire $H$-step sequence rather than a single time point, and it reflects the need to model both temporal dependencies and cross-channel correlations.

Zero-shot forecasting represents the ability of one model to generate predictions for unseen datasets or tasks without any fine-tuning. This is supported by *foundation models*, large pretrained models with significant generalization capacity across tasks, e.g. TimeGPT (Garza & Canseco, 2023), Chronos (Ansari et al., 2024), and Moirai (Woo et al., 2024). Mathematically, zero-shot forecasting can be expressed as $\hat{\boldsymbol{Y}} = \phi(\boldsymbol{X}, H)$ where $\phi$ is a pretrained foundation model. Such models can be directly applied to new forecasting tasks, yielding universal solutions across different domains.

### 3.2 ZERO-SHOT FORECASTING WITH A MODEL ZOO

Despite the broad applicability of individual foundation models, their performance varies with architectures, training methodologies, and inductive biases (Dooley et al., 2023; Gruver et al., 2023). For instance, Chronos demonstrates particular strength in high-frequency electricity data (Aksu et al., 2024), while VisionTS significantly outperforms peers on cloud data with frequent spikes (Toner et al., 2025). This motivates the construction of a *model zoo*: a curated collection of diverse models, denoted as $\mathcal{Z} = \{\phi_1, \phi_2, \ldots, \phi_M\}$. By taking advantage of the complementary properties of different TSFMs, model-zoo forecasting aims to improve accuracy while keeping prediction costs manageable.

The workflow of model zoo forecasting consists of two steps. First, the candidate models are ranked according to their suitability for the input task. Second, predictions are generated by either selecting a single model or aggregating the outputs of the top-ranked models.

**Single Model Selection (Selective Inference).** The zoo first produces a ranking of models based on their estimated zero-shot performance for the input $\boldsymbol{X}$, yielding $\boldsymbol{r}_{\text{final}}$. This allows the system to focus on the most promising candidates rather than evaluating all models.

Table 1: Computational complexity comparison of model selection strategies

| Strategy | Stage | | | Total Complexity per Task |
|---|---|---|---|---|
| | Precompute | Selection | Forecast | |
| Random | – | – | $\mathcal{O}(UN)$ | $\mathcal{O}(N)$ |
| Enumerate all | – | – | $\mathcal{O}(UMN)$ | $\mathcal{O}(MN)$ |
| Ensemble all | – | – | $\mathcal{O}(UMN)$ | $\mathcal{O}(MN)$ |
| Forward-based | – | $\mathcal{O}(UMn)$ | $\mathcal{O}(UN)$ | $\mathcal{O}(Mn + N)$ |
| Repr-based (**Ours**) | $\mathcal{O}(Mn)$ | $\mathcal{O}(U)$ | $\mathcal{O}(UN)$ | $\mathcal{O}(Mn/U + N)$ |

- The complete zero-shot forecasting pipeline consists of at most three stages: (1) *Precompute* model representations if necessary, (2) *Select* suitable models for each task, and (3) *Forecast* using selected models.
- $M$ is the number of models in the zoo, $N$ the dataset size per task, $n$ the subset size for representation construction ($n \ll N$), and $U$ the number of future unseen tasks (assumed large).

**Top-$K$ Ensemble Forecasting (Aggregated Prediction).** Based on this ranking, predictions $\hat{Y}$ are aggregated from the top-$K$ models through averaging. This strategy leverages model diversity to achieve more robust predictions.

$$\boldsymbol{r}_{\text{final}} = \text{Rank}(\boldsymbol{X}, \mathcal{Z}), \quad \hat{\boldsymbol{Y}} = \frac{1}{K} \sum_{m \in \text{TopK}(\boldsymbol{r}_{\text{final}})} \phi_m(\boldsymbol{X}, H). \tag{1}$$

The effectiveness of model zoo forecasting is characterized by two critical metrics: *performance gain* ($\Delta P$) measures the relative improvement in forecasting accuracy compared with the best single model, and *selection efficiency* ($\eta$) quantifies the tradeoff between accuracy and computational cost:

$$\max \begin{cases} \Delta P := 1 - \mathcal{L}(\hat{\boldsymbol{Y}}) \big/ \min_{m=1,\dots,M} \mathcal{L}(\phi_m(\boldsymbol{X}, H)), \\ \eta := \mathbb{E}[\Delta P] \big/ T_{\text{all}}. \end{cases} \tag{2}$$

Here $\mathcal{L}(\cdot)$ denotes the forecasting loss, $\phi_m(\boldsymbol{X}, H)$ is the $H$-step prediction of the $m$-th model, $T_{\text{all}}$ is the total runtime including both model ranking and prediction generation, and $\mathbb{E}[\cdot]$ denotes expectation over tasks. In summary, the model zoo setting enables forecasting through conditional model selection followed by top-$K$ ensemble prediction, with performance evaluated in terms of both accuracy improvement and efficiency.

### 3.3 Challenges with Model Zoo Forecasting

A key challenge in model-zoo-based forecasting is balancing efficiency and accuracy. Contemporary TSFMs often occupy hundreds of megabytes to gigabytes (Chen et al., 2025; Das et al., 2024; Goswami et al., 2024; Woo et al., 2024), making exhaustive evaluation computationally prohibitive. Table 1 summarizes the complexity of different strategies (details in Appendix F.3). (1) *Random* selection offers maximal efficiency but worst accuracy. (2) *Enumerate all* evaluates every TSFM for high accuracy at prohibitive cost. (3) *Ensemble all* averages predictions with similar expense. (4) *Forward-based* methods reduce cost through partial evaluation but still require repeated passes. Although transferability methods (Bao et al., 2019; Deshpande et al., 2021; Nguyen et al., 2020; Pándy et al., 2022; Tan et al., 2021; Tran et al., 2019; You et al., 2021) alleviate some overhead, they remain computationally unaffordable as the zoo expands. This bottleneck mirrors challenges in related domains such as computer vision and natural language processing, and is further compounded by the expensive nature of temporal inference operations. Therefore, it is imperative to develop a lightweight and forward-free model selection mechanism.

Our solution adopts a representation-based (*repr-based*) approach, using "representation" and "embedding" interchangeably. Models and datasets are summarized into embeddings, with model embeddings precomputed once. Suitability is then estimated via efficient similarity matching, amortizing selection cost across tasks. Repr-based selection cuts computational load, scales with zoo size, and offers a transferable suitability measure across datasets.

## 4 ZooCast

Designing a *representation-based* model zoo for time-series forecasting is challenging, as it requires constructing a single unified embedding space that can capture the diverse, non-stationary, and multi-

Figure 2: Overview of ZooCast with **one-embedding-fits-all** design: (a) Advantage subset characterizes model preferences based on inter-model performance analysis. (b) Co-Embedding extractor aligns model and task representations in a shared space for efficient similarity-based model selection. (c) Error-corrected model ranking refines the task prediction by leveraging cross-channel similarity.

channel nature of time series while still enabling fast and accurate model selection. We refer to this as a *one-embedding-fits-all* design: a shared extractor maps both tasks and models into one space so that selection reduces to lightweight similarity matching with near-constant compute. Realizing this goal faces three challenges: (1) identifying model-specific strengths without exhaustive evaluation, because different TSFMs excel on different temporal patterns; (2) embedding tasks and models into a common space despite non-stationarity and multi-channel structure; and (3) turning noisy, per-channel similarities into a stable ranking that is both robust and fast to compute. Our framework addresses these with three coordinated components: advantage subset characterization, model–task co-embedding, and error-correcting consensus ranking. Figure 2 presents the overall design.

## 4.1 Advantage Subset Characterization

A central challenge in building a model zoo is how to characterize the predictive strengths of each TSFM without resorting to exhaustive evaluation. To tackle this, we introduce advantage subset characterization, which identifies model-specific preferences using minimal yet highly discriminative data. As shown in Figure 3, we perform full forward passes of all models on test data, compute prediction variance per sample, and partition samples into variance deciles. The performance gap between the optimal and average models grows exponentially with variance, indicating that high-variance samples carry stronger discriminative signals for model characterization.

Based on this finding, we construct *advantage subsets* that capture each model's unique strengths. To prevent test data leakage, we build a zoo characterization set $\mathcal{D}$ containing $n$ subsequences by randomly sampling source-specific pools $\{\mathcal{D}_i\}$ from the pre-training datasets of each TSFM in the zoo $\{\phi_m\}_{m=1}^M$. The advantage subset for model $\phi_m$ is defined as:

$$\mathcal{D}_m^{\text{adv}} \ = \ \{\, \boldsymbol{x}_i \in \mathcal{D} \mid s_{m,i} \ > \ \tau \,\}. \tag{3}$$

Here, $s_{m,i}$ denotes the *advantage score* and $\tau$ is an adaptive threshold. The detailed computation and interpretation of the *advantage score* are provided in Appendix E.1. This design allows each model's unique strengths to be efficiently profiled while maintaining scalability.

## 4.2 Model-task Representations Co-Embedding

Even with model-specific subsets, another challenge is embedding tasks and models into a shared space despite non-stationarity and multi-channel structure. We address this with a co-embedding extractor that unifies them for efficient similarity-based matching.

### 4.2.1 Build Co-Embedding Extractor

To facilitate efficient model selection, we introduce a general *co-embedding extractor* $\psi : \mathbb{R}^T \to \mathbb{R}^D$, which maps both target time series and foundation models into a shared vector space. The extractor is trained on an independent dataset $\mathcal{D}^*$ (disjoint from both zoo pretraining corpora and downstream forecasting tasks) to prevent information leakage, with an encoder producing latent representations

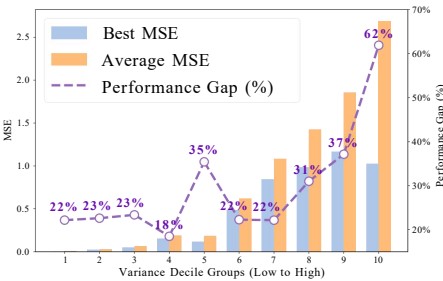

Figure 3: Empirical study: model preference analysis across variance deciles. Samples are drawn from the pre-training datasets of TSFMs, evaluated through full forward passes, and then partitioned into deciles based on the variance of model predictions. The figure illustrates the performance gap in MSE between the optimal and average models within each decile, which shows that high-variance samples are more valuable for characterizing model preferences.

and a decoder reconstructing the original series. The complete training objective integrates three components (detailed formulations are provided in Appendix E.2):

$$\min_{\Theta} \quad \mathcal{L}_{\text{Reconstruction}} + \mathcal{L}_{\text{Constraint}} + \lambda \mathcal{L}_{\text{Transfer}}. \tag{4}$$

Here, $\mathcal{L}_{\text{Reconstruction}}$ encourages temporal fidelity, $\mathcal{L}_{\text{Constraint}}$ applies masked-view contrastive learning to enhance robustness, and $\mathcal{L}_{\text{Transfer}}$ introduces a novel transferability alignment specifically designed for time-series tasks, supervising cross-task similarity with 1-MSE scores. The coefficient $\lambda$ is a balancing factor that controls the relative contribution of the transferability loss to the overall objective. This transferability loss is the core innovation that ensures the learned space reflects actual generalization ability across datasets. Once trained, the extractor provides the foundation for selecting the most suitable models for new tasks, facilitating zero-shot forecasting with high efficiency. The detailed formulations, training setup, and dataset specifications are provided in Appendix E.2.

### 4.2.2 MODEL-TASK REPRESENTATION CONSTRUCTION

For model representation, each foundation model $\phi_m$ is encoded through its advantage subset $\mathcal{D}_m^{\text{adv}}$, where all subsequences are embedded and then averaged into a single vector $\boldsymbol{r}_m$. For the target series $\boldsymbol{X} \in \mathbb{R}^{C \times T}$, task representations are constructed by sampling segments of length $t$ from each channel. The extractor $\psi$ maps these segments into embeddings:

$$\boldsymbol{r}_m = \frac{1}{|\mathcal{D}_m^{\text{adv}}|} \sum_{\boldsymbol{x}_i \in \mathcal{D}_m^{\text{adv}}} \psi(\boldsymbol{x}_i) \in \mathbb{R}^D, \quad m = 1, \dots, M, \qquad \boldsymbol{R}_{\text{zoo}} = [\boldsymbol{r}_1, \dots, \boldsymbol{r}_M]^\top \in \mathbb{R}^{M \times D}.$$

$$\boldsymbol{\mu}_c = \psi(\boldsymbol{x}_c) \in \mathbb{R}^D, \quad c = 1, \dots, C, \qquad \boldsymbol{R}_{\text{task}} = [\boldsymbol{\mu}_1, \dots, \boldsymbol{\mu}_C]^\top \in \mathbb{R}^{C \times D}.$$

Collecting $\boldsymbol{r}_m$ across all $M$ models yields the *model zoo representation library* $\boldsymbol{R}_{\text{zoo}}$, while $\boldsymbol{R}_{\text{task}}$ serves as the task embedding matrix. This symmetric formulation highlights that model representations are averaged over advantage subsets, whereas task representations are constructed channel-wise from sampled segments. Together, they enable efficient similarity computation in a unified vector space, supporting seamless zoo expansion (Appendix F.2) and a lightweight similarity measurement between $\boldsymbol{R}_{\text{zoo}}$ and $\boldsymbol{R}_{\text{task}}$, with $\mathcal{O}(1)$ time complexity for matching operations.

### 4.3 MATCHING TSFMS FOR TARGET TIME-SERIES

Once models and tasks are embedded into a shared space, the key challenge is converting noisy channel-wise similarities into a stable global ranking. We address this with a similarity computation and error-correcting consensus mechanism that aggregates multi-channel signals into robust recommendations for accurate and reliable zero-shot forecasting.

For a target series $\boldsymbol{X}$, we compute the channel–model *similarity score* $\text{sim}_{m,c}$ between each channel embedding $\boldsymbol{\mu}_c \in \mathbb{R}^D$ and each model representation $\boldsymbol{r}_m \in \boldsymbol{R}_{\text{zoo}}$, weighted by a model-specific reliability factor $w_m$ (see Appendix F.1 for details). For each channel, the top-$r$ models ($r = 3$) with the largest $\text{sim}_{m,c}$ are marked in a binary recommendation matrix $\boldsymbol{B} \in \{0,1\}^{C \times M}$, where $B_{c,m} = 1$ indicates model $\phi_m$ is recommended for channel $c$. Aggregating these votes through Hamming distance $h_m$ yields the final ranking $\boldsymbol{r}_{\text{final}}$:

$$\text{sim}_{m,c} = w_m \cdot \frac{\boldsymbol{r}_m^\top \boldsymbol{\mu}_c}{\|\boldsymbol{r}_m\| \|\boldsymbol{\mu}_c\|}, \quad h_m = \sum_{c=1}^{C} \mathbb{I}(B_{c,m} = 0), \quad \boldsymbol{r}_{\text{final}} = \underset{m}{\text{argsort}} \, h_m. \tag{5}$$

This ECOC (Kong & Dietterich, 1995)-inspired approach mimics error correction in digital communications, where cross-channel redundancies mitigate individual channel noise. The final ranking $\boldsymbol{r}_{\text{final}}$ prioritizes models with broad consensus across channels.

Table 2: Aggregated performance on GIFT-Eval, which includes 97 configurations across 23 datasets with diverse domains, prediction lengths, frequencies, and variate counts. We evaluate zero-shot forecasting using 13 TSFMs and their full ensemble (All-13) as basic baselines, compare them against ZooCast-guided (Z.C.) ensembles. Rank is derived from MASE-based ordering across 97 configurations. A lower sMAPE or Rank indicates a better prediction. The best results across each row are **bolded**, while the second best are underlined.

| Metric | Single Model Prediction | | | | | | | | | | | | | Ensemble Prediction | | |
|---|---|---|---|---|---|---|---|---|---|---|---|---|---|---|---|---|
| | Chr.bT | Chr.bM | Chr.bS | Chr.bB | Moi.S | Moi.B | Moi.L | TFM.1 | TFM.2 | Vis.B | Vis.L | Vis.H | Sun.B | All-13 | Top-3 Z.C.(ours) | Top-5 Z.C.(ours) |
| sMAPE | 0.452 | 0.446 | 0.448 | 0.441 | 0.488 | 0.474 | 0.474 | 0.474 | 0.452 | 0.513 | 0.512 | 0.511 | **0.430** | 0.445 | 0.437 | 0.431 |
| Rank | 6.856 | 5.753 | 6.113 | 4.856 | 9.753 | 8.371 | 8.031 | 8.598 | 4.949 | 11.381 | 11.258 | 10.979 | 4.845 | 5.062 | 3.688 | **3.158** |

Table 3: As most existing model selection methods do not support time-series or regression tasks, and LogME was originally designed for time-series forecasting, we adapt its workflow and restrict its application to univariate forecasting, constructing LogME-guided (L.M.) ensembles. The evaluation datasets are reduced to 32 univariate configurations, with all other settings identical to Table 2.

| Metric | Single Model Prediction | | | | | | | | | | | | | Ensemble Prediction | | | | |
|---|---|---|---|---|---|---|---|---|---|---|---|---|---|---|---|---|---|---|
| | Chr.bT | Chr.bM | Chr.bS | Chr.bB | Moi.S | Moi.B | Moi.L | TFM.1 | TFM.2 | Vis.B | Vis.L | Vis.H | Sun.B | All-13 | Top-3 L.M. | Top-5 L.M. | Top-3 Z.C.(ours) | Top-5 Z.C.(ours) |
| sMAPE | 0.369 | 0.363 | 0.358 | 0.354 | 0.401 | 0.383 | 0.383 | 0.391 | 0.389 | 0.460 | 0.458 | 0.457 | 0.356 | 0.364 | 0.377 | 0.369 | **0.352** | 0.353 |
| Rank | 6.531 | 5.438 | 5.250 | 4.562 | 9.625 | 8.719 | 7.656 | 7.531 | 4.875 | 12.031 | 12.219 | 11.906 | 5.562 | 5.000 | 5.375 | 4.969 | 3.688 | **3.094** |

## 5 EXPERIMENTS

We comprehensively evaluate ZooCast through three key analyses: (1) zero-shot point forecasting performance against baseline TSFMs on Gift-Eval (Aksu et al., 2024), a large-scale and best-recognized zero-shot benchmark. (Section 5.1), (2) test-time scaling behavior of ensembles guided by ZooCast's model rankings and ablation studies on critical parameters when constructing the advantage subset in ZooCast (Section 5.2), (3) computational efficiency analysis of repr-based model zoo zero-shot forecasting comparison with forward all TSFMs (Section 5.3). All experiments were conducted under identical hardware and evaluation protocols to ensure fair comparison.

### 5.1 EVALUATION ON GIFT-EVAL BENCHMARK

To assess the generalizability of the model zoo approach, we evaluate ZooCast on GIFT-Eval, a comprehensive benchmark spanning diverse domains and datasets (see Appendix B for full metadata).

**Setup.** Following previous work (Aksu et al., 2024; Liu et al., 2025), we adopt sMAPE as the evaluation metric, excluding CRPS since ZooCast targets point forecasting, whereas CRPS targets probabilistic settings and would bias evaluation of deterministic outputs. Compared to MAPE, sMAPE is symmetric and more robust to small or zero values. We report raw sMAPE values without normalization by a Seasonal Naive baseline. To ensure fair comparison across diverse datasets, we compute a *Rank* score by ranking models within each configuration based on sMAPE and then averaging ranks across all datasets, which mitigates scale differences. Ensemble results are obtained via the naive strategy of simply averaging predictions of the selected TSFMs, highlighting the effectiveness of our framework.

**Baseline.** We evaluate 13 TSFMs as single model prediction baselines: Chronos (Chr.bT/bM/bS/bB) (Ansari et al., 2024), Moirai (Moi.S/B/L) (Woo et al., 2024), VisionTS (Vis.B/L/H) (Chen et al., 2025), TimesFM (TFM.1/2) (Das et al., 2024), and Sundial (Sun.B) (Liu et al., 2025). All models are tested in their original zero-shot configurations. In Table 2 and 3, we report two ensemble baselines: the naive ensemble of all TSFMs (All-13) and LogME-guided ensembles (L.M.), which also serve as our model selection baseline. In Figure 4, we additionally include four classical model selection strategies as baselines: Random Selection, All Current Ensemble, Latest Selection, and Current Best Selection. More experimental settings are provided in Appendix D.

**Real-world Model Selection Evaluation.** We propose a novel evaluation paradigm for sequential model selection, where the model zoo is incrementally updated upon each new TSFM release. In Figure 4, four baseline strategies are implemented: (1) *Random* selection from available models (averaged over 10 seeds), (2) *All Current Ensemble* integrating all released models, (3) *Latest Selection* selecting the most recent release, and (4) *Current Best Selection* choosing historically top-performing models. Since these baselines select or fix models without additional computation, their selection cost is $\mathcal{O}(1)$. This dynamic testing framework validates the model zoo's capability to leverage newly released TSFMs for continuous performance enhancement, demonstrating significant advantages over conventional approaches.

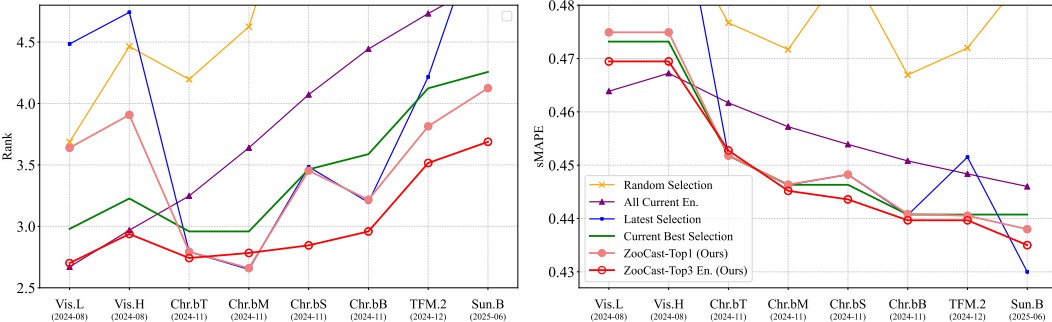

Figure 4: **Real-world model selection with sequential model releases** on GIFT-Eval. The x-axis represents the latest TSFM releases from left to right. Lower Rank values indicate superior relative performance compared to others, while reduced sMAPE reflects improved accuracy. ZooCast (in red) demonstrates consistent superiority across evolving model zoos and unseen datasets, with performance improving as more recent and higher-quality models become available.

**Results.** Table 2 and 3 report aggregated performance across 97 configurations from 23 datasets. ZooCast-guided ensembles consistently outperform all baselines, with **Top-5 Z.C.En.** and **Top-3 Z.C.En.** both delivering highly competitive Rank scores, demonstrating stable average superiority across diverse datasets. Since sMAPE can be affected by extreme values on certain datasets, the clear advantage in Rank provides stronger evidence of overall robustness. Figure 4 further reports performance dynamics under sequential model releases: as more recent TSFMs are added, the sMAPE of Top-3 Z.C.En. steadily improves over time, while its Rank remains consistently the best. This indicates that the larger and more up-to-date the model zoo becomes, the more stable ZooCast's selection advantage is.

## 5.2 ENSEMBLE SCALING BEHAVIOR OF ZOOCAST

The test-time scaling law (Zhang et al., 2025a) refers to the principle that by increasing the prediction stages or computational cost, the prediction performance can improve. A robust scaling strategy should enhance performance with minimal additional computational expense compared to single-model predictions. In ensemble strategies, the optimal approach is to achieve this improvement by accurately selecting a small number of models, avoiding excessive computational costs.

ZooCast provides an effective solution by leveraging model ranks to guide ensemble selection. By choosing only a few top-ranked models, it achieves strong predictive performance at a fraction of the cost of evaluating all candidates. This effect is clearly illustrated in Figure 5, which demonstrates the ensemble scaling behavior as performance steadily improves with a small number of selected models. Furthermore, Figures 5a and 5b validate parameter sensitivity by varying the initial settings (size=1000, threshold=1) from Section 5.1. Across all tested configurations, the Top-3 ensemble consistently outperforms both the naive ensemble (All-13 En.) and the strongest single model (Sun.B), confirming the robustness of ZooCast under different parameter choices.

The results collectively demonstrate ZooCast's unique combination of efficiency and effectiveness: while requiring only minimal computational overhead for model selection, it delivers superior forecasting accuracy that scales gracefully with both ensemble size and model zoo expansion. This dual capability makes ZooCast particularly suitable for production environments where computational resources and prediction quality must be jointly optimized.

## 5.3 EFFICIENCY ANALYSIS

We conduct a computational efficiency analysis comparing our repr-based model zoo approach against forwarding all TSFMs on all 97 forecasting tasks on the Gift-Eval benchmark.

**Setup.** This experiment follows the protocol from the previous section. All evaluations are conducted on the same computing infrastructure, running all three phases across 97 datasets and 13 TSFMs. The complexity analysis adopts the same formulation as in Table 1, with $M = 13$, $N$ varying across tasks, $n = 1000$, and $U = 97$. The precompute stage is completed once per model before any forecasting task arrives, while the selection and forecasting stages are performed per task, thereby incurring no additional per-model cost during inference.

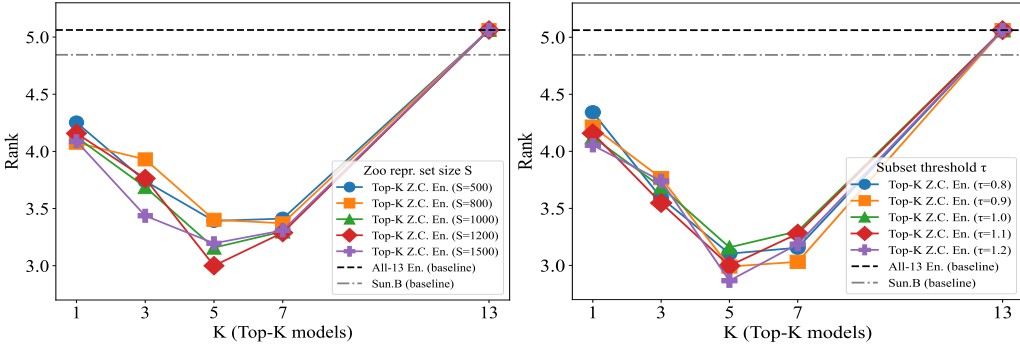

(a) Performance across different repr. set sizes      (b) Performance across different thresholds

Figure 5: **Scaling behavior of ZooCast-guided ensembles.** Lower Rank indicates better performance. (a) and (b) present the performance evaluation across all 97 datasets configurations in GIFT-Eval, analyzing how varying zoo representation set sizes and advantage subset thresholds affect ensemble performance with different ensemble size K. With only a Top-3 ensemble, ZooCast consistently outperforms both the naive ensemble baseline (All-13 En.) and the best single model (Sun.B). These results demonstrate the method's stable test-time scaling behavior, showing consistent performance improvements as the ensemble size lightly expands while maintaining computational efficiency.

Table 4: Efficiency comparison: Full Forward vs. ZooCast (Precompute + Selection + Forecast).

| Stage | Chr.bT | Chr.bM | Chr.bS | Chr.bB | Moi.S | Moi.B | Moi.L | TFM.1 | TFM.2 | Vis.B | Vis.L | Vis.H | Sun.B | Total (s) | Complexity per Task |
|---|---|---|---|---|---|---|---|---|---|---|---|---|---|---|---|
| Full Forward | 974 | 1240 | 1688 | 3445 | 5827 | 10017 | 16141 | 5084 | 7995 | 2930 | 5143 | 8112 | 19428 | **88024** | $\mathcal{O}(MN)$ |
| Precompute | 1.31 | 1.51 | 2.17 | 3.27 | 3.85 | 7.16 | 18.29 | 12.56 | 28.97 | 5.38 | 10.08 | 19.53 | 8.68 | **123** | $\mathcal{O}(Mn/U) \approx \mathcal{O}(1)$ |
| Selection | – | – | – | – | – | – | – | – | – | – | – | – | – | **1042** | $\mathcal{O}(1)$ |
| Forecast | – | – | – | – | – | – | – | – | – | – | – | – | – | **2958** | $\mathcal{O}(N)$ |

**Results.** Table 4 presents the runtime comparison across three execution phases. The *Full Forward* phase simulates a brute-force baseline where all TSFMs are evaluated on every new forecasting task, resulting in a total cost of 88024 seconds. This represents the standard zero-shot approach with no reuse of prior computation. In contrast, the *Precompute* phase incurs a one-time cost of 123 seconds in total for building the representation of each model via forward passes on the zoo representation set $\mathcal{D}$. The *Selection* phase uses the pre-built zoo representation library $\boldsymbol{R}_{\text{zoo}}$ for lightweight model selection, taking 1024 seconds across all tasks. The *Forecast* phase directly generates predictions based on the selection results. Interestingly, we find that the actual speedup achieved by ZooCast is far greater than the number of models in the zoo. This is because the models selected by ZooCast are not only more accurate but also faster to run. While this may partly be coincidental, it suggests that ZooCast tends to favor models that are both superior in performance and more efficient, thereby achieving a dramatic speedup over full forward evaluation.

Specifically, a single matrix computation between $\boldsymbol{R}_{\text{zoo}}$ and the task representation $\boldsymbol{R}_{\text{task}}$ yields the recommendation. This approach enables a constant-time $\mathcal{O}(1)$ process, bypassing the need for any forward computations. By relying on lightweight precomputed representations, we reduce selection complexity from $\mathcal{O}(MN)$ to $\mathcal{O}(1)$ per task, allowing for instant recommendations at inference time and making this repr-based selection highly practical for large-scale zero-shot forecasting. Detailed computational complexity and memory usage analysis is provided in the Appendix F.3.

## 6 CONCLUSION

We present ZooCast, a novel framework for zero-shot time series forecasting that efficiently selects the best model from a zoo of TSFMs. Unlike traditional methods, which rely on single models or exhaustive ensembles, ZooCast reduces model selection to lightweight similarity measurements. Key innovations include the construction of advantage subsets, capturing each model's unique strengths, and a co-embedding strategy for efficient model-task matching. Extensive experiments demonstrate that ZooCast outperforms individual models and naive ensembles in both accuracy and computational efficiency. ZooCast's flexibility allows seamless scalability, enabling the addition of new models without re-evaluating existing ones. As model zoos expand, ZooCast's repr-based selection remains efficient, making it a promising solution for future forecasting tasks.

REPRODUCIBILITY STATEMENT

The results presented in this paper are fully reproducible. Our work focuses on proposing a novel model selection framework for existing time series foundation models (TSFMs), all of which have publicly available implementations and evaluation pipelines documented in the GIFT-Eval benchmark. The minor adaptations we introduce for reproducibility are described in detail in Appendix D. The datasets used in all experiments, along with their preprocessing protocols, are entirely drawn from the publicly available GIFT-Eval benchmark, ensuring transparent and standardized evaluation. For the efficiency analysis, a detailed explanation of the theoretical complexity is provided in Appendix F.3. Moreover, the design of the proposed selection framework, including the advantage subset construction, co-embedding extractor, and consensus ranking mechanism, is fully described in both the main text and the appendix to ensure clarity. To further support reproducibility, the complete source code will be released as supplemental material upon the paper's acceptance.

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

## A MATHEMATICAL NOTATIONS

We summarize the mathematical notations used throughout the paper in Table 5. This convention ensures consistency and aligns with the standardized notation file in the ICLR template.

Table 5: Summary of mathematical notations used in this paper.

| Symbol | Definition |
|---|---|
| $\boldsymbol{X} \in \mathbb{R}^{C \times T}$ | Multivariate input time series with $C$ channels and length $T$ |
| $\boldsymbol{x}_t \in \mathbb{R}^C, \boldsymbol{x}_c \in \mathbb{R}^T$ | Observation at time step $t$; the $c$-th channel sequence of length $T$ |
| $\boldsymbol{Y}, \hat{\boldsymbol{Y}} \in \mathbb{R}^{C \times H}$ | Ground-truth and predicted multivariate sequences of horizon $H$ |
| $\mathcal{Z} = \{\phi_1, \ldots, \phi_M\}$ | Model zoo of $M$ foundation models (TSFMs) |
| $\boldsymbol{r}_{\text{final}}$ | Final model ranking after error-correcting consensus |
| $\Delta P$ | Performance gain relative to the best single model |
| $\eta$ | Selection efficiency, defined as accuracy–cost tradeoff ratio |
| $T_{\text{all}}$ | Total runtime including both model selection and inference |
| $U$ | Count of unseen forecasting tasks to be processed |
| $N$ | Total time-series subsequences to be predicted in a forecasting task |
| $n$ | Subsequences per model for representation construction ($n \ll N$) |
| $\mathcal{D}, \boldsymbol{x}_i$ | Zoo characterization set containing $n$ sampled subsequence $\boldsymbol{x}_i$ |
| $\boldsymbol{E} \in \mathbb{R}^{M \times n}$ | Precomputed MSE/error matrix on $\mathcal{D}$ |
| $s_{m,i}, \tau$ | Advantage score of model $\phi_m$ on sample $\boldsymbol{x}_i$ and its threshold for subset construction |
| $\mathcal{D}_m^{\text{adv}}, d_m, w_m$ | Advantage subset of $\phi_m$, its size, and the derived weight |
| $\sigma_i$ | Inter-model variance of errors on sample $\boldsymbol{x}_i$ |
| $\mathcal{D}^*$ | Independent dataset for training the co-embedding extractor |
| $\psi : \mathbb{R}^T \to \mathbb{R}^D$ | Co-embedding extractor mapping sequences to $D$-dimensional space |
| $\boldsymbol{r}_m, \boldsymbol{\mu}_c \in \mathbb{R}^D$ | Embedding of model $\phi_m$; embedding of channel $c$, $\boldsymbol{\mu}_c = \psi(\boldsymbol{x}_c)$ |
| $\boldsymbol{R}_{\text{zoo}} \in \mathbb{R}^{M \times D}$ | Model zoo representation library of all TSFM embeddings |
| $\boldsymbol{R}_{\text{task}} \in \mathbb{R}^{C \times D}$ | Task representation matrix formed by embeddings of all $C$ channels |
| $\text{sim}_{m,c}$ | Cosine similarity between model $\phi_m$ and channel $c$ |
| $h_m$ | Hamming distance of model $\phi_m$ used in error-correcting consensus ranking |

## B DATASET IN GIFT-EVAL BENCHMARK

**The GIFT-Eval benchmark** (Aksu et al., 2024) is a large-scale, general-purpose evaluation suite designed specifically for testing the zero-shot and universal forecasting capabilities of time series foundation models (TSFMs). The full list of datasets and their metadata is provided in Table 6. GIFT-Eval comprises 23 real-world datasets with 97 configurations collected from diverse sources, encompassing over 144,000 time series and 177 million individual observations. Each dataset is labeled by key structural features, including domain, prediction length, frequency, and number of variates, enabling fine-grained group-wise performance analysis along the following axes:

- **Domain**: Includes seven practical domains: Energy, Economy/Finance, Healthcare, Transport, Nature, Sales, and Web/CloudOps.
- **Prediction Length**: Covering short (6-48), medium (480–600), and long-term (720-900) forecasting challenges.
- **Frequency**: Spanning 10 distinct temporal resolutions, ranging from high-frequency signals—such as 10 seconds (10S), 5 minutes (5T), 10 minutes (10T), and 15 minutes (15T)—to low-frequency records including hourly (H), daily (D), weekly (W), monthly (M), quarterly (Q), and annual (A).
- **Target Variates**: Supporting both univariate and multivariate target structures.

By encompassing both real-world heterogeneity and rigorous statistical structure, GIFT-Eval serves as a challenging and balanced testbed for assessing both accuracy and generalization in zero-shot time series forecasting.

Table 6: Details for all 97 dataset configurations in the GIFT-Eval benchmark.

| Dataset | Domain | Frequency | Variates | Pred Length |
|---|---|---|---|---|
| M4 Hourly | Econ/Fin | H | 1 | 48 |
| M4 Daily | Econ/Fin | D | 1 | 14 |
| M4 Weekly | Econ/Fin | W | 1 | 13 |
| M4 Monthly | Econ/Fin | M | 1 | 18 |
| M4 Quarterly | Econ/Fin | Q | 1 | 8 |
| M4 Yearly | Econ/Fin | A | 1 | 6 |
| ETT1 | Energy | 15T, H | 7 | {48,480,720} |
| ETT1 | Energy | D | 7 | 30 |
| ETT1 | Energy | W | 7 | 8 |
| ETT2 | Energy | 15T, H | 7 | {48,480,720} |
| ETT2 | Energy | D | 7 | 30 |
| ETT2 | Energy | W | 7 | 8 |
| Solar | Energy | 10T, H | 1 | {48,480,720} |
| Solar | Energy | D | 1 | 30 |
| Solar | Energy | W | 1 | 8 |
| Electricity | Energy | 15T | 1 | {48,480,720} |
| Electricity | Energy | H | 1 | {48,480,720} |
| Electricity | Energy | D | 1 | 12 |
| Electricity | Energy | W | 1 | 8 |
| COVID Deaths | Healthcare | D | 1 | 30 |
| US Births | Healthcare | D | 1 | 30 |
| US Births | Healthcare | W | 1 | 8 |
| US Births | Healthcare | M | 1 | 12 |
| Hospital | Healthcare | M | 1 | 12 |
| Jena Weather | Nature | 10T, H | 21 | {48,480,720} |
| Jena Weather | Nature | D | 21 | 30 |
| KDD Cup 2018 | Nature | H | 1 | {48,480,720} |
| KDD Cup 2018 | Nature | D | 1 | 30 |
| Temperature Rain | Nature | D | 1 | 30 |
| Saugeen | Nature | D | 1 | 30 |
| Saugeen | Nature | W | 1 | 8 |
| Saugeen | Nature | M | 1 | 12 |
| Restaurant | Sales | D | 1 | 30 |
| Hierarchical Sales | Sales | D | 1 | 30 |
| Hierarchical Sales | Sales | W | 1 | 8 |
| Car Parts | Sales | M | 1 | 12 |
| Loop Seattle | Transport | 5T, H | 1 | {48,480,720} |
| Loop Seattle | Transport | D | 1 | 30 |
| SZ-Taxi | Transport | 15T | 1 | {48,480,720} |
| SZ-Taxi | Transport | H | 1 | 48 |
| M_DENSE | Transport | H | 1 | {48,480,720} |
| M_DENSE | Transport | D | 1 | 30 |
| BizITObs - Application | Web/CloudOps | 10S | 2 | {60,600,900} |
| BizITObs - Service | Web/CloudOps | 10S | 2 | {60,600,900} |
| BizITObs - L2C | Web/CloudOps | 5T, H | 7 | {48,480,720} |
| Bitbrains - Fast Storage | Web/CloudOps | 5T | 2 | {48,480,720} |
| Bitbrains - Fast Storage | Web/CloudOps | H | 2 | 48 |
| Bitbrains - rnd | Web/CloudOps | 5T | 2 | {48,480,720} |
| Bitbrains - rnd | Web/CloudOps | H | 2 | 48 |

## C  EXTENDED RELATED WORK

We provide additional details on model selection methods relevant to our work.

**Model Selection Methods**. Early model selection in transfer learning relied on brute-force fine-tuning, exhaustively adapting each model to the target task. While accurate, this was infeasible at scale. To reduce cost, *forward-based* methods emerged, approximating model suitability through lightweight inference. LEEP (Nguyen et al., 2020) estimates the conditional distribution between source and target labels via a single forward pass, while LogME (You et al., 2021) computes a marginal likelihood to evaluate feature–label alignment for both classification and regression, though it assumes Gaussian labels and is costly in high dimensions. Despite efficiency gains, forward-based methods still depend on labeled target data and light training, making them incompatible with strict zero-shot settings. To eliminate the need for target labels or forward passes, *representation-based* methods compare tasks in a shared embedding space. TASK2VEC (Achille et al., 2019) pioneered this idea using Fisher information but is tied to classification architectures. TS2Vec (Yue et al., 2022) introduced multi-scale temporal embeddings for time series, but its data-driven embeddings are not aligned with model behavior. SimMTM (Dong et al., 2023) adopted masked modeling to predict missing values, which is lightweight and effective for pretraining but lacks mechanisms for transferability or model selection. TimesURL (Liu & Chen, 2024) enhanced TS2Vec with frequency-time augmentations, yet remains computationally heavy and requires post hoc model ranking. ModelSpider (Zhang et al., 2023) framed model selection as a learning-to-rank task, effective in classification but requiring expensive fine-tuned labels and not extended to regression or time series forecasting. Overall, these methods remain limited by structural rigidity, task specificity, computational overhead, and lack of integrated ranking mechanisms, leaving time series zero-shot selection unresolved.

## D  ZERO-SHOT FORECASTING BENCHMARK DETAILS

This section details the parameter settings used across different stages of the experimental pipeline. All experiments are implemented using PyTorch and executed on a machine equipped with an NVIDIA RTX 4090 GPU and dual AMD EPYC 7763 64-Core CPUs (256 threads in total). To ensure reproducibility, all experiments are conducted with fixed random seeds.

In the Gift-Eval benchmark, the zoo representation set $\mathcal{D}$ contains $n = 1000$ subsequences sampled from the pre-training datasets of TSFMs. We apply the subset thresholding strategy in Equation 6 with an adaptive threshold $\tau = 1$, and define the model weight $w_m$ as the inverse square root of the number of samples in $\mathcal{D}_m^{\text{adv}}$ that exceed the threshold. The co-embedding extractor $\psi$ maps each $\mathbb{R}^{36}$ input segment into a $\mathbb{R}^{128}$ latent representation, resulting in the model zoo representation library $\boldsymbol{R}_{\text{zoo}} \in \mathbb{R}^{M \times 128}$. For task representations, we process the multivariate time series channel-wise: from each channel, five random $\mathbb{R}^{36}$ segments are sampled, encoded with $\psi$, and then averaged to obtain channel embeddings $\boldsymbol{\mu}_c \in \mathbb{R}^{128}$. Stacking across all channels yields the task embedding matrix $\boldsymbol{R}_{\text{task}} = [\mu_1, \mu_2, \dots, \boldsymbol{\mu}_C]^\top \in \mathbb{R}^{C \times 128}$. Model rankings are derived by computing weighted cosine similarities between $\mu$ and $\boldsymbol{R}_{\text{zoo}}$.

**Foundation models as baselines.** For all foundation models, we use publicly available implementations and perform zero-shot evaluation on the test split of each benchmark.

- Chronos (Ansari et al., 2024): We follow the official evaluation setup using `num_samples = 20`. In the GIFT-Eval benchmark, we use four upgraded versions released in November 2024: `chronos-bolt-tiny`, `chronos-bolt-mini`, `chronos-bolt-small`, and `chronos-bolt-base`, denoted respectively as **Chr.bT**, **Chr.bM**, **Chr.bS**, and **Chr.bB**.

- Moirai (Woo et al., 2024): We follow the official evaluation setup using `num_samples = 100` and `patch_size = 32`. In the GIFT-Eval, we set `context_length = 4000`, and `feat_dynamic_real_dim` is set dynamically based on the dataset. We use three different model sizes released in March 2024: `moirai-1.1-R-small`, `moirai-1.1-R-base`, and `moirai-1.1-R-large`, denoted as **Moi.S**, **Moi.B**, and **Moi.L**, respectively.

- TimesFM (Das et al., 2024): We set `context_length = 512` in the GIFT-Eval. We use two different model sizes: `timesfm-1.0-200m` released in March 2024 and

`timesfm-2.0-500m` released in December 2024, denoted as **TFM.1** and **TFM.2**, respectively.

- VisionTS (Chen et al., 2025): Due to the limitations of VisionTS, the maximum context length is 96, while the minimum length depends on the shortest input supported by each dataset. We follow the official implementation and include three model sizes released in August 2024: `mae_visualize_vit_base`, `mae_visualize_vit_large`, and `mae_visualize_vit_huge`, denoted as **Vis.B**, **Vis.L**, and **Vis.H**, respectively.

- Sundial (Liu et al., 2025): We follow the official evaluation setup using `num_samples = 100` and `context_length = 512` in the GIFT-Eval. We use the only public model size: `Sundial_base_128m` released in June 2025, denoted as **Sun.B**.

**Model selection baseline.** We adapted LogME (You et al., 2021), originally designed for CV and NLP tasks, to the time-series forecasting setting by directly applying its core principle of measuring model–data fit. Concretely, we computed LogME scores for each TSFM–task pair and used these scores to generate a model ranking, replacing ZooCast's ranking module. However, this approach faces inherent limitations: it cannot effectively handle multivariate time-series tasks, so our evaluation was restricted to 32 univariate datasets in the GIFT-Eval benchmark. Moreover, LogME is computationally inefficient, since it requires forward passes of all TSFMs for every new task and does not allow reusing precomputed results, leading to substantial overhead compared with our design.

**Ground Truth Evaluation.** In real zero-shot scenarios, true model rankings are unknown. To validate our method, we first conduct an exhaustive evaluation of all models in $\mathcal{M}$ across all test sets to establish reference metrics. This comprehensive evaluation constitutes the benchmark for assessing ZooCast's recommendation accuracy.

# E    ADDITIONAL METHODOLOGICAL DETAILS

This section provides the extended technical details that complement the main text. Specifically, we present (i) the formulation of advantage scores for constructing model-specific subsets, (ii) the full design of the co-embedding extractor with its loss functions, training setup, and datasets.

## E.1    ADVANTAGE SCORE COMPUTATION

We first forward all models on the zoo characterization set $\mathcal{D}$ to obtain the MSE matrix $\boldsymbol{E} \in \mathbb{R}^{M \times n}$, where $E_{m,i}$ is the MSE of $\phi_m$ on subsequence $\boldsymbol{x}_i$. Let the inter-model error variance on sample $\boldsymbol{x}_i$ be $\sigma_i = \mathrm{std}\big(\{E_{m,i}\}_{m=1}^{M}\big)$, and let $\bar{\sigma}$ and $\hat{\sigma}$ be the mean and standard deviation of $\{\sigma_i\}$. The per-sample *advantage score* for model $\phi_m$ is then defined as

$$s_{m,i} = \left( \frac{1}{M-1} \sum_{k \neq m} E_{k,i} - E_{m,i} \right) \cdot \frac{\sigma_i - \bar{\sigma}}{\hat{\sigma}}, \quad \mathcal{D}_m^{\mathrm{adv}} = \{ \boldsymbol{x}_i \in \mathcal{D} \mid s_{m,i} > \tau \}. \quad (6)$$

This score jointly reflects (i) how much better model $\phi_m$ performs compared to its peers on sample $\boldsymbol{x}_i$, and (ii) how discriminative the sample is based on inter-model disagreement. A sample is included in $\mathcal{D}_m^{\mathrm{adv}}$ if $s_{m,i} > \tau$, where $\tau$ is an adaptive threshold. This design ensures that advantage subsets focus on high-variance, informative samples that best reveal model-specific strengths. The cardinality $d_m = \big| \mathcal{D}_m^{\mathrm{adv}} \big|$ is used to derive the model weight $w_m$ in selection (details in Appendix F.1).

## E.2    CO-EMBEDDING EXTRACTOR DETAILS

For completeness, we describe the full design of the co-embedding extractor, including all loss formulations (in equation 4), parameter configurations, and training data details.

**Loss functions:**

$$\mathcal{L}_{\text{Reconstruction}} = \min_{\Theta} \ \|\boldsymbol{x} - \hat{\boldsymbol{x}}\|_2^2, \quad \boldsymbol{x} \in \mathcal{D}^*,$$

$$\mathcal{L}_{\text{Constraint}} = -\sum_{\boldsymbol{x} \in \mathcal{D}^*} \sum_{\tilde{\boldsymbol{x}} \in \mathcal{M}(\boldsymbol{x})} \log \frac{\exp(\text{sim}(\psi(\boldsymbol{x}), \psi(\tilde{\boldsymbol{x}})))}{\sum_{\boldsymbol{x}' \in \mathcal{D}^*} \exp(\text{sim}(\psi(\boldsymbol{x}), \psi(\boldsymbol{x}')))},$$

$$\mathcal{L}_{\text{Transfer}} = \sum_{\boldsymbol{x}_i \in \mathcal{D}_i, \ \boldsymbol{x}_j \in \mathcal{D}_j} \left\| g_{i,j} - \text{sim}\big(\psi(\boldsymbol{x}_i), \psi(\boldsymbol{x}_j)\big) \right\|_2^2.$$

Here, $\boldsymbol{x} \in \mathbb{R}^T$ is a subsequence sampled from the auxiliary training set $\mathcal{D}^*$, and $\hat{\boldsymbol{x}}$ is its reconstruction. $\psi(\boldsymbol{x})$ denotes the embedding representation of input sequence $\boldsymbol{x}$. The operator $\mathcal{M}(\boldsymbol{x})$ denotes masked versions of $\boldsymbol{x}$ obtained by randomly dropping time points along the temporal axis, ensuring consistency between $\psi(\boldsymbol{x})$ and $\psi(\tilde{\boldsymbol{x}})$. The similarity operator $\text{sim}(\cdot, \cdot)$ denotes cosine similarity. The transferability score is defined as $g_{i,j} = 1 - \text{MSE}(\phi_i, \mathcal{D}_j)$, where $\phi_i$ is a foundation model pretrained on dataset $\mathcal{D}_i$, and $\text{MSE}(\phi_i, \mathcal{D}_j)$ measures its forecasting error on dataset $\mathcal{D}_j$.

The three losses play complementary roles: (1) $\mathcal{L}_{\text{Reconstruction}}$ is a standard encoder–decoder objective ensuring temporal fidelity of representations, (2) $\mathcal{L}_{\text{Constraint}}$ employs masked-view contrastive learning along the temporal dimension to enhance robustness against short-term noise while capturing long-term dynamics, and (3) $\mathcal{L}_{\text{Transfer}}$ introduces a novel supervised alignment mechanism tailored for time-series foundation model selection. To the best of our knowledge, this is the first work to propose such a mechanism in this setting.

**Training setup.** We train an encoder–decoder extractor with input length 36, prediction length 12, patch size 16, one encoder layer, and hidden dimension 64. Training runs for 10 epochs using MSE loss with a learning rate of 0.001.

**Training Datasets.** Approximately 300,000 subsequences are sampled from M3, M5, and Tourism datasets to compute cross-task transferability scores and to construct advantage subsets for unseen tasks. In contrast, Moirai pretrains on LOTSA (2.7B points) and Sundial on TimeBench (1T points), highlighting that our extractor relies on relatively modest-scale data with short subsequences (length 36) to ensure computational efficiency.

# F   FURTHER INFORMATION ON ZOOCAST

This section supplements ZooCast's architectural design by formally explaining why the framework achieves high *accuracy*, *scalability*, and *efficiency*. Through detailed mathematical formulations, we clarify how each design decision contributes to the system's overall effectiveness.

## F.1   ACCURACY: EMBEDDING-AWARE MODEL WEIGHTING

During the *precompute* stage of ZooCast, we not only construct the model zoo representation library $\boldsymbol{R}_{\text{zoo}}$, but also record a weighting factor $w_m$ for each model $\phi_m$ that reflects its relative strength. This design is motivated by a key observation: although state-of-the-art (SOTA) models continue to evolve, at any given time, some models significantly outperform others across the majority of tasks. Treating all models equally in the selection phase would dilute this prior knowledge and underutilize the generalizability of strong models.

But how can we determine which models in the zoo are superior when the downstream task is unknown? We determine model weights during the construction of the *advantage subset* $\mathcal{D}_m^{\text{adv}}$ as described in Appendix E.1, where $\tau$ is an adaptive threshold. The number of samples that satisfy the threshold condition for model $\phi_m$ is denoted by $d_m$:

$$d_m = \left| \mathcal{D}_m^{\text{adv}} \right|, \quad w_m = \frac{1}{\sqrt{d_m}}$$

We then define the model weight as the inverse square root of $d_m$, ensuring that models with stronger performance (i.e., higher $d_m$) receive lower weights. These weights are incorporated in the selection

phase to modulate the similarity scores between tasks and models. A lower weight effectively shortens the computed distance between a task and the corresponding model, increasing the likelihood that the model is prioritized during selection.

During selection, these weights are used to modulate similarity scores (equation 5), which favors models that demonstrated stronger performance during the precompute phase, even when their raw embedding similarity is slightly lower.

## F.2 SCALABILITY: SEAMLESS MODEL ZOO EXPANSION

ZooCast enables rapid and scalable integration of new models through an efficient update mechanism that avoids redundant computation. Assume the current model zoo consists of $M$ models, for which the full evaluation matrix $\boldsymbol{E}^{\text{old}} \in \mathbb{R}^{M \times n}$ has been precomputed (provided in Appendix E.1), where $n$ denotes the number of time series samples in the fixed zoo representation set $\mathcal{D}$. Correspondingly, the model zoo representation library is maintained as $\boldsymbol{R}_{\text{zoo}}^{\text{old}} \in \mathbb{R}^{M \times D}$, where $D$ is the dimensionality of the representation vector for each semantic unit.

When a new model $\phi_{M+1}$ is added, the characterization process completes in under one minute. Specifically, the model is full-forwarded on the fixed dataset $\mathcal{D}$ to obtain its MSE vector and latent representations, which are used to append a new row to both the error matrix and the zoo representation library. Crucially, this update only requires inference for the new model and does not involve re-evaluating any existing models, thereby keeping the computation cost constant regardless of the current zoo size. The transformation of the key components is illustrated below:

$$\boldsymbol{E}^{\text{old}} \in \mathbb{R}^{M \times n} \xrightarrow{\text{Add } \phi_{M+1} \text{ results}} \boldsymbol{E}^{\text{new}} \in \mathbb{R}^{(M+1) \times n} \quad (< 1 \text{ min})$$

$$\boldsymbol{R}_{\text{zoo}}^{\text{old}} \in \mathbb{R}^{M \times D} \xrightarrow{\text{Append new repr.}} \boldsymbol{R}_{\text{zoo}}^{\text{new}} \in \mathbb{R}^{(M+1) \times D} \quad (< 1 \text{ s})$$

This update procedure is repeated for every new model without increasing the overall computational burden as the zoo grows, since the time cost is dominated by the one-time forward pass of the incoming model on $\mathcal{D}$. Under this "precompute-and-select" paradigm, ZooCast achieves seamless and scalable model zoo expansion with negligible overhead, ensuring long-term adaptability to evolving foundation model landscapes.

## F.3 EFFICIENCY: MINIMAL COMPUTATIONAL COMPLEXITY AND MEMORY USAGE

This subsection explains how different model selection strategies lead to the computational complexities shown in Table 1, and why ZooCast achieves extreme efficiency in continuous model selection, reducing the per-task cost from $\mathcal{O}(MN)$ to $\mathcal{O}(Mn/U + N)$ for unseen prediction tasks.

**Random selection.** The simplest baseline is to randomly choose a single model without any selection procedure. In this case, the selection cost is negligible, and only the chosen model needs to be run on the $N$ data points of the target task. The total complexity is therefore $\mathcal{O}(N)$ per task. This represents an achievable upper bound on efficiency but typically results in poor accuracy.

**Enumerate all.** A brute-force strategy is to forward all $M$ models on every task and then select the best model afterward. This requires $\mathcal{O}(UMN)$ operations across all tasks, or $\mathcal{O}(MN)$ per task. While accurate, this method becomes prohibitively expensive as $M$ grows.

**Ensemble all.** Another naive approach is to ensemble the predictions of all models without selection. This also requires forwarding all $M$ models on each task, leading to the same complexity as the brute-force method, $\mathcal{O}(MN)$ per task. In practice, the gain in accuracy is often small compared to the computational overhead.

**Forward-based selection.** Several transferability-based approaches (e.g., LogME, LEEP) estimate model suitability by forwarding each candidate model on part of the task data. If each model uses $n \ll N$ samples to compute its score, the selection phase costs $\mathcal{O}(Mn)$ per task, and the subsequent forecasting phase costs $\mathcal{O}(N)$. The total per-task complexity is $\mathcal{O}(Mn+N)$. Although more efficient than brute force, this cost still scales linearly with $M$ and remains high for large model zoos.

**Repr-based (ZooCast).** Our method introduces a *precompute-and-select* paradigm that fundamentally changes the scaling behavior of model zoo inference. In the precompute phase, each of the $M$ models is forwarded on a small subset of $n$ samples to build its representation, yielding a one-time cost of $\mathcal{O}(Mn)$. This cost is paid only once when the zoo is constructed, and never repeated for subsequent tasks. When averaged over $U$ downstream tasks, the per-task amortized cost is $\mathcal{O}(Mn/U)$, which converges to $\mathcal{O}(1)$ as $U$ grows large with fixed $n \ll N$. In the selection phase, instead of re-running models, ZooCast only performs cosine similarity between the task representation $\boldsymbol{R}_{\text{task}} \in \mathbb{R}^{C \times D}$ and the precomputed model zoo embeddings $\boldsymbol{R}_{\text{zoo}} \in \mathbb{R}^{M \times D}$. These are low-dimensional ($D \leq 128$) vector operations, so all computations reduce to matrix slicing, dot products, and max-pooling. Each of these costs constant time per model, and with parallelization, the effective cost per task is $\mathcal{O}(1)$. This sharp contrast with forward-based approaches highlights the scalability of ZooCast: regardless of the zoo size $M$, the selection stage does not grow in complexity. Finally, in the forecast phase, ZooCast generates predictions only with the top-selected models, requiring $\mathcal{O}(N)$ operations per task. Thus, the entire pipeline reduces the total per-task cost from $\mathcal{O}(Mn + N)$ in forward-based methods to $\mathcal{O}(Mn/U + 1 + N)$, which approaches $\mathcal{O}(N)$ for large $U$.

**Memory usage (ZooCast).** ZooCast's memory footprint is dominated by (1) the precomputed error matrix $\boldsymbol{E} \in \mathbb{R}^{M \times n}$ and the zoo representation library $\boldsymbol{R}_{\text{zoo}} \in \mathbb{R}^{M \times D}$. Both scale linearly with $M$. In practice, at $M = 10$, storage is only about 0.2 MB, and even at $M = 1000$, storage remains manageable (around 20 MB).

In conclusion, random selection is efficient but inaccurate, brute-force and ensemble-all are accurate but computationally prohibitive, and forward-based methods provide a compromise but still scale poorly with zoo size. ZooCast, by contrast, achieves near-optimal efficiency: the precompute phase amortizes to $\mathcal{O}(1)$ per task, the selection phase is also $\mathcal{O}(1)$ thanks to lightweight similarity computation, and only the forecast phase scales with the task length at $\mathcal{O}(N)$. Together, this makes the total per-task complexity $\mathcal{O}(1 + 1 + N) \approx \mathcal{O}(N)$, which matches the theoretical efficiency upper bound while retaining strong accuracy. With extremely low memory consumption, ZooCast makes the vision of *One-Embedding-Fits-All* practical for real-time, large-scale zero-shot forecasting.

# G  DATASET USAGE SUMMARY

We summarize the three types of datasets used in this work and clarify collection effort and leakage safeguards.

**(1) Auxiliary training set for the co-embedding extractor** ($\mathcal{D}^*$). We sample approximately 300,000 subsequences (length 36) from public time-series corpora (M3, M5, Tourism) to train the extractor $\psi$ and to supervise the transferability loss (Appendix E.2). This set is *fully disjoint* from both TSFM pretraining corpora and downstream evaluation data. Its scale is moderate by design (short segments, standard datasets), which keeps training efficient while providing broad pattern diversity. Once $\psi$ is trained, $\mathcal{D}^*$ is not accessed again during the precompute or evaluation phases, preventing any backdoor leakage into selection or forecasting.

**(2) Zoo characterization set for advantage subsets and model embeddings** ($\mathcal{D}$). For each TSFM $\phi_m$ in the zoo, we randomly draw only *hundreds* of subsequences from the model's *public* pretraining pools (as released by the original TSFM papers, e.g., Chronos, Moirai). This small, diverse set is used *once* to compute the error matrix $\boldsymbol{E} \in \mathbb{R}^{M \times n}$, construct the advantage subsets $\mathcal{D}_m^{\text{adv}}$, and build the model zoo representation library $\boldsymbol{R}_{\text{zoo}}$ (Appendix E.1). Collection is straightforward (public sources, random sampling), and because $\mathcal{D}$ comes from TSFM pretraining data rather than downstream benchmarks, there is *no* leakage path into the evaluation sets. After precompute, $\boldsymbol{R}_{\text{zoo}}$ is fixed and reused across tasks; neither labels nor raw series from $\mathcal{D}$ are needed at test time.

**(3) Downstream evaluation data** (GIFT-Eval). All zero-shot results are reported on the public GIFT-Eval benchmark (23 datasets, 97 configurations; Appendix B). GIFT-Eval is used *only* for evaluation and for producing model rankings at test time via lightweight similarity between $\boldsymbol{R}_{\text{task}}$ and $\boldsymbol{R}_{\text{zoo}}$; it is never used to train $\psi$ or to construct $\boldsymbol{R}_{\text{zoo}}$. Thus, evaluation happens under a strict zero-shot protocol with clear phase isolation.

**Leakage safeguards and phase isolation.** Data usage follows a clean separation: $\mathcal{D}^*$ is used exclusively to train $\psi$; $\mathcal{D}$ (public TSFM pretraining pools with only small random samples) is used exclusively to precompute $\boldsymbol{E}$, $\mathcal{D}_m^{\text{adv}}$, and $\boldsymbol{R}_{\text{zoo}}$; GIFT-Eval is reserved exclusively for downstream

evaluation. No dataset is reused across phases in a way that could leak target information into training or precompute. Moreover, our method requires *diversity but not scale*: the precompute step relies on small, randomly sampled and publicly documented sources, which further reduces collection burden and eliminates hidden data dependencies. Together, these choices ensure simple collection, transparency, and *no leakage risk* across all stages.

## H   LIMITATIONS

While ZooCast introduces a novel and efficient framework for zero-shot model selection in time-series forecasting, several limitations remain. The paradigm of using a model zoo for prediction, though increasingly common in vision and language domains, is still nascent in the time-series community. Consequently, wider adoption of such frameworks may require additional time for ecosystem development, tooling, interpretability improvements, and community validation. Furthermore, the effectiveness of our method currently depends on a curated zoo and a fixed representation extractor trained on pre-defined datasets, which may not fully capture the diversity of future task distributions. In addition, the model selection process itself is somewhat constrained: the extractor only supports fixed-length inputs (length 36) and does not yet accommodate dynamic sequence lengths or multimodal/co-variable information from domains such as text or images. The integration scheme employed in our current design is also relatively simplistic, discarding potentially valuable multi-quantile prediction information and lacking mechanisms for dynamic task adaptation, thereby limiting ensemble expressiveness. Finally, the performance ceiling of ZooCast is inherently bounded by the strongest individual TSFM in the zoo; when integration is too basic or ensembles are not employed, the method cannot consistently surpass the best single model. These limitations highlight important avenues for future work, including more flexible representation learning, richer integration strategies, and adaptive ensembles that can better exploit the strengths of diverse TSFMs.

## I   THE USE OF LARGE LANGUAGE MODELS

In preparing this manuscript, we employed large language models (LLMs) primarily as auxiliary tools for non-scientific tasks, including language polishing, table formatting, and mathematical notation checking. These tools were used to improve clarity and presentation consistency without influencing the research design, experimental results, or scientific conclusions. All technical content, algorithms, and experiments remain the original contributions of the authors.

