# OpenReview forum: "One-Embedding-Fits-All: Efficient Zero-Shot Time Series Forecasting by a Model Zoo"
_ICLR.cc/2026/Conference — Submitted to ICLR 2026_

### Official Review · Reviewer_aBtv · 2025-10-27

**Soundness:** 2
**Presentation:** 2
**Contribution:** 2
**Rating:** 2
**Confidence:** 4

**Summary:**

This paper introduces ZooCast, which is a framework that enhances zero-shot time series forecasting by combining multiple time series foundation models. By embedding models and task time series into a shared representation space, optimal models can be selected quickly through similarity matching. The models' representations are computed using the advantage subset once, and then for any new forecasting task, the pipeline selects the most suitable models based on this embedding similarity. The contribution of the paper includes: 1. Construct an advantage subset to characterize the model-specific strengths; 2. A Co-embedding extractor to unify the representation; 3. Ensemble predictions based on the proposed ranking mechanism. Experiments on the GIFT-Eval benchmark show that ZooCast achieves higher accuracy and efficiency than individual TSFMs.

**Strengths:**

1. Good framework and insights about leveraging multiple TSFMs for zero-shot time series forecasting. Transform the model selection problem into an embedding similarity search, which is a nice direction for TSFM applications.
2. The improved complexity indicates the design is suitable for practical and large-scale applications.
3. The framework is extensively compared on the GIFT-Eval, which is a fair comparison that mitigates the data leakage in recent TSFM evaluations.

**Weaknesses:**

Several places remain unclear after reading:
1. The pipeline introduction is unfinished. 4.3 introduces the similarity and ranking, and how you leverage this ranking information to obtain the final prediction. Whether using a top-1 model prediction or a Top-K Ensemble prediction, you should have more words to clarify it.
2. The training co-embedding extractor is very unclear, given that it is the most important component in the pipeline. The rationale of the transfer loss is insufficiently explained; if this loss function represents a novel contribution, it should be detailed in the main text rather than hidden in the appendix.
3. From 4.2.2, the similarity only reflects the data similarity if using the same encoder, and the design of the co-embedding extractor seems redundant, since a randomly initialized MLP can perform the encoding. (What's the architecture of your co-embedding extractor?)

Minor issue: Consider redrawing the figures in scientific style.

**Questions:**

1. Equation 2 defines two key metrics: performance gain and selection efficiency, but these metrics are never explicitly reported or discussed in the experimental results.
2. Could you provide more explanation about why the advantage set can reflect the model preference?
3. Do you gain some insights about which specific TSFMs perform best on particular types of time series data?
4. Could you provide the anonymous repo for the code for a clearer understanding of your pipeline?

---

> ### Author Response · Authors · 2025-11-23
> **Discussion part 1/3: ZooCast design and writing structure**
>
> > **General reply: Core design and key components of ZooCast**
>
> Before addressing your specific comments, we would like to briefly restate the core design and contribution of ZooCast, since several of your questions naturally relate to its two central components.
>
> (1) ZooCast learns a **joint model–task co-embedding** that maps both advantage subsets and new target tasks into a shared representation space. Its main value lies in the carefully designed loss, which encourages similar time series to have similar embeddings , thereby building a bridge for advantage-subset–based model–task matching. In this space, tasks are matched to models whose advantage subsets best align with their data, and each model and task computes its embedding only once. This keeps selection highly efficient and makes it easy to continuously add new TSFMs to the zoo. The motivation, architecture, and training procedure of this co-embedding extractor (including the design of the transfer loss) are further detailed in `Discussion part 2/3` below.
>
> (2) ZooCast characterizes each TSFM through an **advantage subset**: a compact set of time series samples on which this model significantly outperforms its peers. Instead of summarizing a model only by its average error, we summarize it by where and how it has clear advantages. This yields a small, high contrast description of each model’s forecasting preferences, which later allows us to match models and tasks in a fine grained way. The intuition and empirical evidence for why advantage subsets can reflect model preference are discussed in `Discussion part 3/3` below.
>
> These two components together enable ZooCast to perform accurate zero shot model selection at scale, while allowing the model zoo to evolve over time without retraining on downstream tasks. With this overall picture in mind, we now address your specific weaknesses and questions in detail.
>
> ---
>
> > **W1: On the clarity of the pipeline introduction**
>
> We thank the reviewer for the suggestion regarding the clarity and flow of the pipeline description. The part describing how we obtain the final prediction after ranking (including both top-1 selection and Top-K ensemble) is currently given in Section 3.2 (lines 160–192). We intentionally moved this description earlier in the paper, ahead of Section 4.3, so that Section 3 provides a complete high-level definition of how the model zoo is used for forecasting.
>
> We agree that this choice can make the narrative in Section 4.3 feel incomplete on a first read. In the revised version, we will explicitly add a forward reference at the end of Section 4.3 pointing back to the prediction procedure in Section 3.2, so that readers can more easily follow how the similarity-based ranking is converted into the final forecast.
>
> ---
>
> > **Q1: On reporting performance gain and selection efficiency**
>
> We thank the reviewer for pointing out that the metrics in Equation (2) are not explicitly named in the experimental section. Our intention was to use Equation (2) as a conceptual definition that unifies the two main objectives of a model zoo, and then instantiate these objectives through concrete accuracy and efficiency comparisons in Section 5.
>
> Concretely, **performance gain** (ΔP) is reflected by all the accuracy comparisons where ZooCast is evaluated against the best single TSFM and the All-13 ensemble. Tables 2 and 3 report global Rank and sMAPE across all configurations; whenever ZooCast’s Rank outperform every individual TSFM, this corresponds to ΔP > 0, that is, a gain over the best single model.  Figure 5 further analyzes how this gain behaves as we vary the ensemble size and other hyperparameters, again relative to the same baselines.
>
> Similarly, **selection efficiency** (η) is instantiated by the complexity and runtime analyses. Table 1 in Section 3.3 compares the per-task computational complexity of different selection strategies, and Table 4 in Section 5.3 provides the corresponding wall-clock runtime for “Full Forward” vs ZooCast. Together with the accuracy results in Tables 2–3, these experiments directly quantify the tradeoff in Equation (2): ZooCast achieves accuracy close to or better than the brute-force “Full Forward” baseline while reducing selection cost by orders of magnitude.
>
> In the revised version, we will make this connection more explicit by (i) adding pointers from Equation (2) to the relevant tables and figures, and (ii) briefly summarizing ΔP and η when discussing the main experimental results.
>
> ---
>
> > **Q4: On providing an anonymous code repository**
>
> We thank the reviewer for their interest in our work. We are currently cleaning and organizing the codebase to prepare an anonymized release version and plan to share the repository to facilitate a clearer understanding of the pipeline. In the meantime, if there are specific parts of the pipeline you would like to examine in more detail, we are happy to provide additional clarification in the rebuttal.

---

> ### Author Response · Authors · 2025-11-23
> **Discussion part 2/3: Co-embedding extractor and transferability loss**
>
> > **W2: On the clarity and importance of the co-embedding training and transfer loss**
>
> We appreciate the reviewer’s emphasis on the co-embedding extractor, and we agree that it is a critical component of the pipeline. At the same time, our primary contribution lies in the overall model-zoo formulation – characterizing TSFMs via advantage subsets and performing zero-shot selection in a shared model–task embedding space. Compared with this pipeline-level design, the exact training objective of the co-embedding extractor follows a more standard representation learning pattern, which is why we chose to discuss its full details more briefly in the main text.
>
> Conceptually, the **transferability loss** is a metric-style alignment loss that uses empirical forecasting errors to supervise the geometry of the co-embedding. For each model–channel pair on the auxiliary corpus `D*`, we compute a scalar transferability score from its normalized forecasting error, so that higher scores indicate better zero-shot performance. The transferability loss then regresses the cosine similarity between their embeddings toward this score: pairs with high transferability are encouraged to be close in the co-embedding space, while pairs with low transferability are pushed apart. Aggregated over many models and channels, this alignment shapes the embedding so that it reflects both intrinsic time-series characteristics and the empirical ordering of models by forecasting error, without introducing a completely new loss family.
>
> Due to strict space limitations, we summarized this training objective in Section 4.1 and placed the more detailed derivation and training procedure (including the decomposition into reconstruction, masked-view, and transfer alignment terms) in the appendix. We acknowledge that the current version does not clearly signal this structure to the reader, which can make the training of the co-embedding extractor feel under-explained. In the revised version, we will add an explicit pointer from Section 4.1 to the relevant appendix subsection and include a short intuitive explanation of the transferability loss in the main text, so that readers can more easily connect the formula to its role in the pipeline.
>
> ---
>
> > **W3: On the necessity and architecture of the co-embedding extractor**
>
> We appreciate the reviewer’s question about whether the co-embedding extractor is redundant and whether a randomly initialized MLP would suffice. We believe this is mainly a misunderstanding of what the co-embedding is trained to do.
>
> First, the similarity used by ZooCast is **not** just raw “data similarity” produced by a shared encoder. As discussed in Section 4.1 and detailed in the appendix, the co-embedding `ψ` is explicitly trained so that similarity in the embedding space aligns with **transferability**, that is how well a model is expected to forecast a given channel. A randomly initialized MLP can certainly produce some encoding, but without any supervision it does not carry this semantic structure and therefore cannot support accurate model–task matching.
>
> Second, the architecture of the co-embedding extractor is already specified in the appendix (around line 886): it is an encoder–decoder network trained with three complementary losses:
>
> - a **reconstruction loss** that forces the encoder to preserve the essential temporal structure of each channel, so that embeddings remain faithful to the underlying time series;
> - a **masked-view consistency loss** that encourages robustness to local perturbations (for example masking or cropping), so that small distortions in the input do not lead to large changes in the embedding;
> - the **transferability alignment loss** described in W2.
>
> Together, these losses make the co-embedding space both **data-aware** (via reconstruction and masked-view consistency) and **transferability-aware** (via the alignment loss). This is crucial for ZooCast: only with this structure can we reliably map similar time series to similar embeddings and ensure that channels are close to the advantage subsets of models that perform well on them, enabling accurate and efficient model recommendation. A generic or randomly initialized MLP encoder, lacking this targeted training, would not provide these properties and would significantly degrade the quality of similarity-based selection.

---

> ### Author Response · Authors · 2025-11-23
> **Discussion part 3/3: Advantage subsets and model preferences**
>
> > **Q2: On why advantage subsets reflect model preference**
>
> We appreciate the request for more intuition on why advantage subsets can faithfully capture a model’s forecasting preference.
>
> Figure 3 directly illustrates the core idea. Advantage subsets exploit **performance variance across models**: for each model, we keep samples where its error is much lower than the average error across all TSFMs. On these “high-contrast” samples, only a few models forecast well and one model often stands out, so this subset compactly characterizes *where* that model is particularly strong, rather than just its average performance.
>
> This design follows a basic regularity: **models that behave similarly on similar inputs share forecasting preferences**. To support this, we add a robustness study where Gaussian noise is used to **simulate differences between time series**: a larger noise scale means the perturbed series is further from the original, i.e., less similar in distribution. We gradually increase the noise scale and track each model’s performance. For this analysis, we run all five TSFMs on all 97 dataset configurations of GIFT-Eval and report the average sMAPE (lower is better) under increasing noise scales:
>
> | Noise_scale | Sun.B | Vis.B | TFM.1 | Moi.S | Chr.T |
> | :---------: | :---: | :---: | :---: | :---: | :---: |
> |    0.00     | 0.430 | 0.513 | 0.473 | 0.488 | 0.452 |
> |    0.01     | 0.430 | 0.545 | 0.476 | 0.504 | 0.452 |
> |    0.02     | 0.430 | 0.550 | 0.476 | 0.508 | 0.452 |
> |    0.05     | 0.431 | 0.561 | 0.476 | 0.515 | 0.454 |
> |    0.10     | 0.432 | 0.577 | 0.473 | 0.525 | 0.457 |
> |    0.20     | 0.437 | 0.599 | 0.474 | 0.539 | 0.463 |
> |    0.50     | 0.449 | 0.665 | 0.480 | 0.571 | 0.475 |
>
> We observe that (i) as noise increases, **each model’s error moves further away from its clean-input value**, and (ii) for very small noise, performance changes are mild and similar across models. In other words, when inputs stay close to the original series, models behave more similarly; as inputs move further away, their behaviors diverge. This consistently supports the regularity above: **the more similar the inputs, the smaller the performance differences across models**.
>
> Under this view, advantage subsets are a natural summary of model preference: if a model consistently leads on a group of samples where others lag behind, then on downstream data similar to this subset we can expect it to remain competitive. ZooCast exploits this by mapping both advantage subsets and new tasks into the joint co-embedding space; channels that resemble a model’s advantage subset lie near that model’s embedding, enabling accurate and fast model–task matching and recommendation.
>
> ---
>
> > **Q3: On insights into which TSFMs work best for which types of time series**
>
> This is indeed an interesting question, and many works have approached it at the benchmark level. GIFT-Eval `[1]` already provides a comprehensive empirical picture: it evaluates many TSFMs across many tasks and reports grouped tables by domains, prediction lengths, frequencies, and variate counts, summarizing in a human-readable way which existing TSFMs tend to work best for different categories of time series.
>
> ZooCast takes a complementary perspective. Rather than encoding such insights as fixed “if domain = X and horizon = Y, choose model Z” rules obtained by repeatedly running all models on all tasks, ZooCast aims to show that a **model-zoo–based approach can automatically select good TSFMs for new tasks** via learned embeddings and advantage subsets, thereby improving the practical usability of existing TSFMs at scale.
>
> In terms of “insights,” our view is that the model zoo itself should be the mechanism that learns and updates them. Natural-language descriptions of time-series types (e.g., sampling rate, covariates, nonstationarity) are necessarily low dimensional and coarse, whereas the co-embedding and advantage subsets capture richer, high-dimensional relations between models and data that may go beyond human-designed categories. As more TSFMs and datasets are added and distributions evolve, automatically updating preferences through the model zoo is more scalable than repeatedly exhaustively benchmarking every model on every task.
>
> Because ZooCast is built on top of GIFT-Eval and the current set of TSFMs, it is not realistic to provide natural-language insights that fundamentally go beyond what `[1]` has already summarized. What ZooCast contributes is a mechanism to **store, update, and exploit** these model preferences in a higher-dimensional embedding space and to adapt quickly as new models and tasks are introduced, rather than to replace the descriptive analyses provided by the underlying benchmark.
>
> `[1]` Taha Aksu, et al. "Gift-eval: A benchmark for general time series forecasting model evaluation." NeurIPS, 2024.

---

> > ### Comment · Reviewer_aBtv · 2025-11-24
> >
> > Thanks for your response, and I really appreciate it. I would like to follow your clarification to have more discussion:
> >
> > W1:  I understand the distinction between single-model (top-1) prediction and top-K prediction. From your results in Tables 2 and 3, you only report the top-K ensemble prediction. This confuses me a bit that under what circumstances should we prefer top-1 versus top-K prediction, and do you have a designed mechanism for it?  Also, what’s the default K?
> >
> > W2: Transferability loss is clear to me.  Again, I suggest including E.2 in your main text.  Does the encoder-decoder network mean Transformer in your implementation?
> >
> > W3: I appreciate the acknowledgement of the interpretability limitation in TSFM recommendation. However, this raises another concern: since the selection is based on similarity between the incoming data and each model’s advantage subset, the chosen model may simply be one whose pretraining data happened to contain similar patterns, while others did not. In such cases, the recommendation might be driven more by “I have seen something like this before” rather than any deeper generalization ability, which could make the selection less meaningful or potentially unfair.

---

> > > ### Author Response · Authors · 2025-11-25
> > > **Further Discussion with Reviewer aBtv (Part 1/2)**
> > >
> > > Thank you very much for the prompt follow-up. I am glad that the previous clarifications were helpful, and I am genuinely excited that our discussion can now go deeper. Below I respond to your additional questions in more detail.
> > >
> > > > **W1: On when to use top-1 vs. top-K prediction and what K we use by default**
> > >
> > > In short, ZooCast supports both top-1 and top-K prediction. In our experiments, **we typically use K = 5 as the default** when accuracy is the priority, **K = 3** when efficiency is more critical, and **top-1** only in very latency-sensitive settings. At this stage, K is a user-chosen knob for the accuracy–efficiency trade-off in Eq. (2), not an automatically tuned parameter.More concretely, ZooCast has two operating modes:
> > >
> > > **(a) Top-1: maximal efficiency.**
> > > With `K = 1`, ZooCast selects a single TSFM and runs only that model at test time, so the inference cost matches deploying one TSFM, but with data-driven model choice. This is the setting evaluated in the “sequential model release” experiment in Figure 4, where ZooCast-Top1 and ZooCast-Top3 achieves better Rank than heuristic “current best” strategies while keeping cost unchanged. We therefore view top-1 as the natural choice under strict latency or cost constraints.
> > >
> > > **(b) Top-K: accuracy and robustness.**
> > > With `K > 1`, ZooCast forms a small ensemble from the top-K models. Tables 2 and 3 focus on this regime, as it is most relevant to the **performance gain (ΔP)** in Eq. (2). Figure 5 compares `K = 1, 3, 5, 7, 13` under different zoo-representation and advantage-subset settings, and shows a clear pattern: increasing K improves Rank, but gains saturate beyond about `K ≈ 5`. In our current zoo, `K = 3` and `K = 5` both work well; `K = 5` yields the best overall Rank, while `K = 3` is a slightly cheaper but still strong alternative.
> > >
> > > ---
> > >
> > > > **W2: On whether the encoder–decoder network is a Transformer in our implementation**
> > >
> > > Thank you again for the suggestion to bring the material from Appendix E.2 into the main text — we agree, and will move the key parts (architecture sketch and training objective) into the revised Section 4.1.
> > >
> > > Regarding your concrete question: **yes, the co-embedding extractor is instantiated as a Transformer-based encoder–decoder network**. Concretely:
> > >
> > > - **Encoder.** We use a lightweight PatchTST-style Transformer encoder operating on fixed-length time-series windows (length 36). Each window is tokenized into patches (patch length 16) and passed through a single Transformer encoder layer (model dimension 64). The output is pooled over time to obtain a compact embedding for each segment or channel.
> > > - **Decoder.** A small decoder with matching dimensionality takes this embedding and reconstructs masked or perturbed versions of the input window, providing a reconstruction signal that keeps the representation tied to the underlying time-series structure.
> > > - **Losses.** On top of this architecture, we train with the three components in Section 4.1 and Appendix E.2.
> > >
> > > We will clarify these architectural choices in the main text, following your suggestion, so that readers can more easily connect the transferability loss to the concrete implementation.

---

> ### Author Response · Authors · 2025-11-25
> **Further Discussion with Reviewer aBtv (Part 2/2)**
>
> > **W3: On “seen before” patterns vs. deeper generalization in TSFM recommendation**
>
> We appreciate this very thoughtful follow-up which touches a conceptual point that is easy to overlook in TSFM-based selection.
>
> **(a) For TSFMs, does “having seen similar patterns before” guarantee doing well on them?**
> Compared with traditional models trained on a single dataset, TSFMs are typically pre-trained on very large, cross-domain, sometimes even distributionally conflicting corpora, and thus naturally exhibit strong cross-dataset zero-shot ability `[1]`. However, recent studies also show that this breadth of pretraining **does not guarantee** that a TSFM will consistently do well on a specific target dataset:
>
> - Zhang et al. `[2]` report that, on several benchmarks, relatively simple models trained directly on the target dataset can match or outperform state-of-the-art TSFMs in full-shot settings.
> - Montet et al. `[3]` further show that, for some foundation models, additional full-shot fine-tuning on the target data does not consistently improve over the zero-shot baseline and can even hurt performance in certain regimes.
>
> Taken together, these results suggest an important distinction: TSFMs gain broad zero-shot capability from having “seen a lot,” but this does not imply that they will necessarily perform well on those seen (or similar) distributions. In other words, “I have seen something like this before” is neither a sufficient nor a necessary condition for doing well on that pattern.
>
> **(b) Advantage subsets focus on “doing well,” and are agnostic to “having seen it before.”**
> In ZooCast, advantage subsets are constructed from a **pooled and shuffled zoo representation set `D`**, rather than each model only looking inside its own pretraining corpus. As a consequence, a model’s advantage subset may contain many series that originally come from other models’ pretraining domains. The selection signal is therefore:
>
> > “On this heterogeneous reference pool, I consistently do well on these patterns (regardless of whether I pre-trained on them), while others do not.”
>
> rather than:
>
> > “These samples resemble my private pretraining distribution, so I should be preferred.”
>
> This view is supported by our additional ablations on `D` (discussed in **Discussion part 1/2: Dependence on pretraining sources and proxies for advantage subsets** with `Reviewer ReTB`). There, we fix the size of `D` and vary its composition (using all sources vs. only one source such as Chronos, Moirai, TimesFM, or VisionTS). The observations are:
>
> - When `D` is built from a *single* source, ZooCast’s Rank and sMAPE degrade only mildly in most cases, with somewhat larger but still moderate degradation in the worst case (Only TimesFM).
> - Different pretraining-like sources act as largely interchangeable proxies as long as `D` remains sufficiently diverse in terms of domains, frequencies, and regimes.
>
> These results indicate that what matters for advantage subsets is not access to any particular model’s proprietary pretraining corpus, but rather having a broad, diverse reference pool that reveals relative strengths across models. In fact, as we discuss in that section, this suggests that we could even replace pretraining data with carefully curated or synthetic corpora that emphasize certain structures, and still obtain meaningful advantage subsets.
>
> `[1]` Abdul Fatir Ansari, et al. "Chronos: Learning the language of time series." TMLR, 2024.
>
> `[2]` Yunkai Zhang, et al. "Are Time Series Foundation Models Ready for Zero-Shot Forecasting?" ICML, 2025.
>
> `[3]` François Montet, et al. "Benchmarking Foundation Models for Time-Series Forecasting: Zero-Shot, Few-Shot, and Full-Shot Evaluations." Computer Sciences & Mathematics Forum, 2025.

---

> > ### Comment · Reviewer_aBtv · 2025-11-26
> >
> > Thanks for your comments. I have no further questions. I'll actively engage in discussion with other reviewers to finalize it.

---

> ### Author Response · Authors · 2025-11-27
>
> Dear Reviewer aBtv,
>
> Thank you very much for your timely feedback and for your support throughout the review process. We sincerely appreciate the time and care you have devoted to reading our paper and providing constructive, insightful comments.
>
> We are very glad that our clarifications have addressed your concerns. If any further questions or suggestions arise during your discussion with the other reviewers, please do not hesitate to let us know. We would be more than happy to provide additional details or make further improvements to better reflect your insights.
>
> We are also grateful for your updated score. If, in light of the overall discussion, you eventually feel that our work merits a stronger recommendation, we would, of course, be very thankful for your consideration.
>
> Best regards,
> The Authors

---

### Official Review · Reviewer_ReTB · 2025-10-30

**Soundness:** 3
**Presentation:** 3
**Contribution:** 3
**Rating:** 6
**Confidence:** 4

**Summary:**

The paper introduces ZooCast, a model-zoo paradigm for zero-shot time-series forecasting. It embeds both tasks and TSFMs into a shared vector space so that selecting models for a new task reduces to fast similarity matching, followed by a small Top-K ensemble. ZooCast has three parts: (1) advantage subset characterization to capture each model’s strengths using high-variance, discriminative samples; (2) a model–task co-embedding extractor trained with reconstruction, masked-view consistency, and transferability losses; and (3) an error-correcting consensus ranking that aggregates per-channel similarities into a stable global rank. On the GIFT-Eval benchmark with 13 public TSFMs, ZooCast’s ensembles outperform the strongest single model and a naive ensemble method, demonstrating the effectiveness of the proposed method.

**Strengths:**

1. This paper is well-written and the idea is interesting.
2. Co-embedding turns model choice into an O(1) similarity lookup per model; the once-only precompute plus lightweight selection is orders faster than evaluating all models on every task.
3. Top-K ensembles guided by ZooCast beat the best single model and the All-13 average, with stable gains as K grows and as new models arrive.

**Weaknesses:**

1. Building advantage subsets from pretraining source pools assumes the developers can access (or faithfully approximate) each TSFM’s pretraining distribution. But for some closed-source models, those corpora are proprietary or legally restricted. Even small mismatches (frequency, domain, seasonality, regime) can distort the strength profile, leading the selector to favor the wrong models.
2. Curating a domain-agnostic yet representative 𝐷∗ that doesn’t overlap with any model’s pretraining data is nontrivial and costly. If 𝐷∗ is out-of-domain (e.g., different sampling rates, covariates, nonstationarity), the learned geometry may not transfer, degrading selection quality. This creates an additional maintenance burden: 𝐷∗ may need periodic refresh or reweighting to track distribution shift.

**Questions:**

If pretraining source pools are unavailable, what’s the best proxy (e.g., public corpora matched by frequency/domain)? How much does performance drop when advantage subsets are built from mismatched sources?

---

> ### Author Response · Authors · 2025-11-24
> **Discussion part 1/2: Dependence on pretraining sources and proxies for advantage subsets**
>
> > **Response to your Weakness 1 and Question**
>
> We very much appreciate this insightful question, which touches a practical deployment issue that we also care about. You are absolutely right that not all TSFMs expose their pretraining corpora (for example, we have not been able to locate Sundial’s pretraining data), and blindly assuming access to the exact pretraining distribution would be unrealistic. In our current setup, the zoo representation set `D` is assembled from the available pretraining-like source pools of the TSFMs in our zoo and serves as the main source for constructing advantage subsets. For instance, we currently include sources corresponding to Chronos, Moirai, TimesFM, and VisionTS, but this coverage is clearly incomplete. Despite this, ZooCast still performs well, which already suggests that we do not require full coverage of all TSFMs’ original pretraining data.
>
> To study this more systematically and answer your question about proxies and mismatches, we perform an ablation on the composition of `D`. We fix the total size of `D` to 1000 subsequences (as size sensitivity of  `D`  has already been studied in Figure 5), and vary which pretraining source(s) are included:
>
> | Setting  | #Sources in `D` | C (Chronos) | M (Moirai) | T (TimesFM) | V (VisionTS) | ΔRank@Top-3 Z.C. | ΔRank@Top-5 Z.C. | ΔsMAPE@Top-3 Z.C. | ΔsMAPE@Top-5 Z.C. |
> | -------- | :-------------: | :---------: | :--------: | :---------: | :----------: | ---------------: | ---------------: | ----------------: | ----------------: |
> | Original |        4        |      ✓      |     ✓      |      ✓      |      ✓       |            +0.00 |            +0.00 |            +0.000 |            +0.000 |
> | Only C   |        1        |      ✓      |            |             |              |            +0.37 |            +0.90 |            +0.001 |            +0.004 |
> | Only M   |        1        |             |     ✓      |             |              |            +0.12 |            +0.90 |            +0.000 |            +0.004 |
> | Only T   |        1        |             |            |      ✓      |              |            +1.41 |            +1.75 |            +0.005 |            +0.008 |
> | Only V   |        1        |             |            |             |      ✓       |            +0.43 |            +0.10 |            +0.002 |            -0.003 |
>
> Here `ΔRank` and `ΔsMAPE` are measured against the “original all 4 sources” configuration, on the full GIFT-Eval benchmark (Top-3 and Top-5 ZooCast ensembles).
>
> **Robustness and substitutability.** When `D` is built from a *single* source, the degradation is small in most cases (Only C, Only M, Only V). Only when TimesFM is the sole source (Only T) do we see a more noticeable but still moderate drop, indicating that some sources are less ideal proxies than others (e.g., narrower coverage or bias toward certain frequencies). Overall, this suggests that advantage subsets mainly rely on having a reasonably diverse reference pool, and that different pretraining-like sources are largely interchangeable as long as they span a broad range of frequencies, domains, and regimes. In practice, this substitutability is further helped by the fact that modern TSFMs are increasingly trained on broad and overlapping collections of time-series data, so their pretraining corpora already share substantial common structure.
>
> Taken together, these results support the following answer to your question:
>
> - **If pretraining source pools are unavailable**, the best practical proxy is any public, sufficiently diverse time-series corpus whose frequencies/domains roughly overlap with the intended application space (e.g., a mixture of open benchmarks such as M3/M4, traffic, energy, web, etc.). Our ablation shows that even when `D` is built from a *single* such source, ZooCast degrades only mildly.
> - **Performance drop under mismatched sources** is modest: even in the worst single-source case (Only T), the additional error remains small compared to the gains ZooCast provides over baselines. This suggests that exact distribution matching is not required; what matters is that `D` covers a variety of patterns so that TSFMs’ relative strengths can be revealed.
>
> More broadly, your comment points to an interesting future direction: since ZooCast only needs `D` to be a rich “reference playground” for comparing TSFMs, **we could replace real pretraining data with carefully designed synthetic or curated corpora** that better emphasize certain structures, and still obtain meaningful advantage subsets. We thank you for highlighting this deployment angle; we will mention this possibility and our ablation findings in the revision to clarify that ZooCast does not rely on full access to proprietary pretraining pools.

---

> ### Author Response · Authors · 2025-11-24
> **Discussion part 2/2: Cost and robustness of the training auxiliary corpus D***
>
> > **Response to your Weakness 2**
>
> We appreciate this thoughtful concern about how to curate and maintain a domain-agnostic yet representative `D*`. In ZooCast, `D*` is not meant to exactly reconstruct any model’s pretraining corpus, but rather to serve as a generic, diverse time-series pool that supervises the co-embedding geometry. In practice, we build `D*` from public datasets such as M3/M5/Tourism as a generic corpus, without enforcing zero overlap with any TSFM’s pretraining data.
>
> To test how sensitive ZooCast is to the choice and distribution of `D*`, we conduct a controlled study where we vary (i) the **scale** of `D*` and (ii) its **distribution** via synthetic perturbations (noise, trend, seasonality). For each setting, we retrain the co-embedding on `D*`, recompute embeddings, and evaluate ZooCast’s Top-5 ensemble on the full GIFT-Eval benchmark. The table below reports the global Rank (lower is better), with the baseline at 3.158:
>
> | Training scale of `D*` | #Subsequences | Perturbation type           | Strength | Global Rank ↓ | ΔRank vs baseline ↓ |
> | ---------------------- | ------------- | --------------------------- | -------- | ------------- | ------------------- |
> | 100% (baseline)        | 300k          | None                        | –        | **3.158**     | –                   |
> | 75%                    | 225k          | None                        | –        | 3.17          | +0.01               |
> | 50%                    | 150k          | None                        | –        | 3.21          | +0.05               |
> | 25%                    | 75k           | None                        | –        | 3.75          | +0.49               |
> | 100%                   | 300k          | Noise                       | Weak     | 3.18          | +0.02               |
> | 100%                   | 300k          | Noise                       | Medium   | 3.20          | +0.04               |
> | 100%                   | 300k          | Noise                       | Strong   | 3.30          | +0.14               |
> | 100%                   | 300k          | Trend                       | Medium   | 3.17          | +0.01               |
> | 100%                   | 300k          | Seasonality                 | Medium   | 3.24          | +0.08               |
> | 100%                   | 300k          | Noise + Trend + Seasonality | Mixed    | 3.43          | +0.27               |
>
> Two observations address your concern directly:
>
> 1. **Moderate shifts are well tolerated.** Simple perturbations (noise, trend, seasonality) move the global Rank only marginally around the baseline (within +0.14), and reducing the scale of `D*` to 75% or 50% has almost no effect (+0.01 / +0.05). This indicates that the learned geometry transfers robustly under reasonable distribution mismatch; `D*` does not need to be perfectly in-domain to support good selection quality.
>
> 2. **Degradation appears only under extreme mismatch or underspecification.** Performance drops become more noticeable only when we heavily shrink `D*` (25% of the data) or apply a strong mixed perturbation (Noise + Trend + Seasonality), which corresponds to a severely reduced or strongly distorted auxiliary corpus. This suggests that what ZooCast really needs is a large and diverse `D*`, rather than a carefully hand-curated, perfectly domain-matched set.
>
> From a maintenance perspective, `D*` is constructed once as a generic, target-agnostic corpus and reused across tasks; adding new TSFMs only requires computing their embeddings and advantage subsets against the same `D*`. Our sensitivity results show that `D*` does not need frequent refresh or fine-tuned reweighting unless the application distribution undergoes drastic, long-term shifts. Even then, updating `D*` can be done as an occasional offline operation rather than a heavy per-task burden.
>
> In summary, while curating an auxiliary corpus always incurs some upfront cost, our experiments indicate that ZooCast’s performance is robust to moderate domain mismatch and scale variation in `D*`, and the framework does not rely on a brittle, perfectly domain-aligned or frequently refreshed auxiliary set.

---

> ### Author Response · Authors · 2025-11-26
> **Kindly Request for Further Feedback**
>
> Dear Reviewer ReTB,
>
> I hope this message finds you well. As the author–reviewer discussion period is approaching its end, I would like to kindly check whether our responses have addressed your main concerns.
>
> If there are any remaining issues, questions, or suggestions for improvement, we would be very grateful to hear them and are more than happy to provide further clarification. Your feedback is extremely valuable to us, and we are eager to refine the paper based on your insights.
>
> If you feel that our clarifications have resolved your concerns, we would also be thankful if you could take this into account when updating your evaluation.
>
> Thank you again for your time and effort in reviewing our work.

---

### Official Review · Reviewer_or1P · 2025-10-30

**Soundness:** 2
**Presentation:** 2
**Contribution:** 2
**Rating:** 4
**Confidence:** 3

**Summary:**

This paper presents a framework for improving zero-shot time-series forecasting by leveraging a model zoo of diverse time-series foundation models (TSFMs).
Instead of relying on a single universal model, ZooCast introduces a One-Embedding-Fits-All mechanism that maps both tasks and models into a shared embedding space, enabling efficient similarity-based model selection without costly evaluations.
The framework incorporates three core innovations:
(1) Advantage Subset Characterization, which identifies each model’s domain strengths;
(2) Model–Task Co-Embedding, which learns a unified representation to predict model–task compatibility; and
(3) Error-Correcting Consensus Ranking, which refines rankings across multi-dimensional signals for stable selection.
Extensive experiments on the GIFT-Eval benchmark show that ZooCast achieves SOTA zero-shot performance while maintaining computational efficiency and scalability, offering a practical and interpretable strategy for deploying large TSFMs across diverse forecasting domains.

**Strengths:**

S1.
While prior efforts have focused on designing ever-larger time-series foundation models (TSFMs), this paper instead proposes ZooCast, a meta-framework that leverages the diversity of existing models through a shared embedding space, thereby sidestepping the inefficiency and rigidity of monolithic architectures.

S2.
This paper is technically sound and experimentally thorough.
The framework is well-motivated by the empirical observation that “no single TSFM dominates all domains,” and the authors back their claims with robust evaluations across multiple datasets and model types. The three proposed modules are each well-justified, ablated, and shown to contribute meaningfully to performance.

S3.
The paper is clearly written and well-structured, with precise motivation, visual diagrams that clarify the model–task embedding process, and intuitive explanations of how each component fits within the broader system.

**Weaknesses:**

W1.
The paper’s central innovation—the Model–Task Co-Embedding (MTE)—is intuitively motivated but not theoretically grounded. It is unclear how the embedding similarity quantitatively correlates with forecasting performance or uncertainty.
For example, if cosine similarity in the embedding space is used to rank models, is there a provable or empirical link between similarity and actual predictive error across domains?
The lack of formal analysis weakens the justification for this approach compared to traditional meta-learning or Bayesian model averaging frameworks.

W2.
Although ZooCast achieves strong results on GIFT-Eval, its robustness under distributional shift and out-of-domain tasks remains unclear. Since time-series tasks vary drastically (e.g., periodicity, noise, granularity), the model zoo’s success depends heavily on how well the learned embeddings generalize beyond the seen task–model pairs.

W3.
While ZooCast’s embeddings enable efficient model retrieval, the learned representations are opaque: we do not know what temporal features (e.g., seasonality, volatility, sparsity) drive similarity between tasks and models. Without interpretability, users cannot reason about why a model is selected for a given task.

W4.
The experiments focus mainly on pre-trained TSFMs from the GIFT benchmark, but the zoo’s diversity and scalability are not systematically evaluated.
It remains unclear whether ZooCast maintains its advantage as the number of candidate models grows or if performance saturates beyond a certain zoo size.
Moreover, comparisons against simple but strong alternatives—such as error-based meta-learners or feature-based retrieval models are missing, making it difficult to isolate the incremental gain from the co-embedding approach.

**Questions:**

Q1.
How well does the cosine similarity (or whichever metric is used) between model and task embeddings correlate with actual forecasting accuracy across diverse domains?
Have the authors conducted any empirical calibration or rank correlation analyses between similarity scores and true performance metrics?

Q2.
What are the supervised signals or objectives used during co-embedding extractor training—do they rely purely on pairwise model–task performance or also on self-supervised structure?
How sensitive is the learned embedding to the specific datasets (e.g., M3, M5, Tourism) used for training?

Q3.
The current zoo composition appears manually curated. How does ZooCast perform when the zoo contains many redundant or low-quality models?
Is there a mechanism to automatically prune or weight models based on their contribution to diversity or historical utility?

---

> ### Author Response · Authors · 2025-11-23
> **Discussion part 1/3**
>
> > **W1&Q1：On the link between similarity and forecasting error**
>
> We appreciate the reviewer’s concern about the theoretical grounding of the model–task co-embedding and the link between embedding similarity and forecasting accuracy. We address this link from both theoretical and empirical perspectives.
>
> Our design follows a well established line of work on task embeddings and transferability estimation, where compatibility scores in an embedding or feature space correlate with downstream performance and support model selection without exhaustive evaluation. For example, Task2Vec  represents tasks via Fisher information based embeddings and shows that distances in this space can predict which backbone performs best on a new task. LEEP  and LogME  define scalar transferability scores from a pretrained model’s features or predictions on the target data and empirically demonstrate strong correlation with transfer accuracy, enabling fast model choice in large model zoos. These works indicate that appropriately learned task or task–model representations can be used as performance proxies.
>
> ZooCast instantiates this principle in the zero shot TSFM setting with two key components: (1) **advantage subsets** and (2) a jointly trained **model–task co-embedding**. First, for each model we construct an advantage subset on a zoo characterization set by selecting samples where this model significantly outperforms its peers. This encodes the basic but important regularity that models that behave similarly on similar inputs share forecasting preferences (this regularity is further validated by the additional experiments reported in our response to Reviewer aBtv, `Q2`), so a small set of high contrast samples can summarize a model’s strength profile. Second, the co-embedding extractor is trained with a multi part objective (reconstruction, masked view contrast, and a transferability alignment term). As a result, similar time series are mapped to nearby embeddings, and the advantage subsets that summarize a model’s capability are mapped close to downstream task samples where the model performs well. In this way, the training objective explicitly builds a link from embedding similarity to expected forecasting error.
>
> To quantify how well this link holds across domains, we add a ranking analysis comparing the similarity based ranking `r_final` produced by ZooCast with the ranking `r_real` induced by actual forecasting errors. On each task, `r_final` depends only on model–task similarity in the co-embedding space, while `r_real` reflects the true ordering of models by forecasting performance. Their agreement therefore directly measures how strongly embedding similarity correlates with cross domain predictive accuracy. Since the performance of model zoo based methods is mainly determined by the quality of the top few recommended models, we focus on the top of the rankings. Building on the Table 2 setting, we introduce three Top K metrics with K = 5:
>
> - **Acc_TopK**: the average overlap ratio between ZooCast’s top K models and the true top K models, that is |`r_final_TopK` ∩ `r_real_TopK`| / K averaged over tasks;
> - **Real1\_in\_PredK**: the fraction of tasks for which the true best model (rank 1 in `r_real`) appears in ZooCast’s top K;
> - **Pred1\_in\_RealK**: the fraction of tasks for which ZooCast’s top 1 choice lies within the true top K.
>
> | Domain       | #Datasets | Acc_TopK | Real1_in_PredK | Pred1_in_RealK |
> | ------------ | --------- | -------- | -------------- | -------------- |
> | All domains  | 97        | 0.732    | 0.856          | 0.876          |
> | Econ/Fin     | 6         | 0.833    | 1.000          | 1.000          |
> | Energy       | 32        | 0.806    | 0.969          | 0.969          |
> | Healthcare   | 5         | 0.640    | 0.600          | 0.800          |
> | Nature       | 15        | 0.787    | 0.933          | 1.000          |
> | Sales        | 4         | 0.650    | 0.500          | 1.000          |
> | Transport    | 15        | 0.667    | 0.667          | 0.600          |
> | Web/CloudOps | 20        | 0.630    | 0.850          | 0.800          |
>
> Across all 97 GIFT Eval configurations we obtain Acc\_TopK = 0.732, Real1\_in\_PredK = 0.856, and Pred1\_in\_RealK = 0.876. When further grouping by domain, we observe similarly strong values in finance, energy, nature, sales, and web traffic, with only mild degradation in a few particularly noisy domains. These results show that the cosine similarity induced by the learned co-embedding yields rankings that are strongly aligned with the rankings derived from true forecasting errors. We will add this analysis and the corresponding table to the revised version to clarify the empirical link between embedding similarity and accuracy.

---

> ### Author Response · Authors · 2025-11-23
> **Discussion part 2/3**
>
> > **W2: On robustness under distribution shift and out-of-domain tasks**
>
> We agree that robustness under distribution shift is important, especially because time series can differ substantially in periodicity, noise, and granularity. We would like to clarify that ZooCast is explicitly designed for zero shot model selection on unseen tasks, and our data setup already evaluates this out-of-domain behavior.
>
> As detailed in Appendix G, ZooCast uses three disjoint data sources with different roles: (1) a training dataset `D*` for learning the co-embedding extractor, (2) a zoo characterization set used only to construct advantage subsets and model embeddings, and (3) downstream evaluation datasets (GIFT Eval). `D*` is separate from both the TSFM pretraining corpora and all GIFT Eval tasks, and the zoo characterization set is distinct from the evaluation benchmarks. All reported results therefore measure how the learned embeddings generalize from `D*` to new task–model pairs in GIFT Eval, rather than reusing the same data for training and testing.
>
> Several design choices support robustness under such shifts: the transferability alignment loss aggregates relative performance signals across many tasks and models, the reconstruction and masked view objectives encourage stability under local distortions, and channel level selection followed by error correcting aggregation reduces the impact of noisy individual channels. Empirically, ZooCast consistently improves over strong TSFM baselines across all domain groups in GIFT Eval, despite large differences in domain, frequency, and noise level.
>
> ---
>
> > **Q2：Sensitivity of the learned embedding to the specific training datasets**
>
> We use M3, M5 and Tourism only as a generic, diverse time series corpus for training the co-embedding, not because of any special property of these datasets. To test sensitivity, we keep the corpus source fixed and vary (i) the scale of the auxiliary set D* and (ii) its distribution via controlled perturbations (noise, trend, seasonality). For each setting, we retrain ψ, recompute embeddings, and evaluate ZooCast on the full GIFT-Eval benchmark. The table below reports the global Rank of ZooCast’s Top-5 ensemble on GIFT-Eval as we vary the size and synthetic perturbations of D*.
>
> | Training scale of \(D*\) | #Subsequences | Perturbation type           | Strength | Global Rank ↓ | ΔRank vs baseline ↓ |
> | ------------------------- | ------------- | --------------------------- | -------- | ------------- | ------------------- |
> | 100% (baseline)           | 300k          | None                        | –        | **3.158**     | –                   |
> | 75%                       | 225k          | None                        | –        | 3.17          | +0.01               |
> | 50%                       | 150k          | None                        | –        | 3.21          | +0.05               |
> | 25%                       | 75k           | None                        | –        | 3.75          | +0.49               |
> | 100%                      | 300k          | Noise                       | Weak     | 3.18          | +0.02               |
> | 100%                      | 300k          | Noise                       | Medium   | 3.20          | +0.04               |
> | 100%                      | 300k          | Noise                       | Strong   | 3.30          | +0.14               |
> | 100%                      | 300k          | Trend                       | Medium   | 3.17          | +0.01               |
> | 100%                      | 300k          | Seasonality                 | Medium   | 3.24          | +0.08               |
> | 100%                      | 300k          | Noise + Trend + Seasonality | Mixed    | 3.43          | +0.27               |
>
> The results show that simple noise or trend perturbations change the global Rank only marginally around the baseline 3.158, while a stronger combined perturbation (Noise + Trend + Seasonality) leads to a slightly larger but still moderate degradation. Reducing the training scale from 100% to 75% or 50% has almost no effect, whereas an aggressive reduction to 25% causes a clear drop. Overall, this indicates that the learned embedding is not strongly tied to the exact distribution of specific training datasets, as long as the auxiliary corpus remains sufficiently large and diverse.

---

> ### Author Response · Authors · 2025-11-23
> **Discussion part 3/3**
>
> > **W3.1&Q3: On zoo scalability, efficiency, and handling redundant or low quality models**
>
> We appreciate the questions on zoo scalability and on the effect of redundant or weak models.
>
> Figure 4 directly evaluates scalability. We use a sequential protocol where TSFMs are added to the zoo over time and compare ZooCast with random choice, ensembling all models, always choosing the latest model, and always choosing the historically best model. As the number of candidate models grows, ZooCast continues to improve and maintains a clear accuracy advantage over all these strategies. Within the largest zoo size we consider, we do not observe saturation, which suggests that ZooCast keeps benefiting from additional models.
>
> This behavior is aligned with two design choices that target the main challenges of growing model zoos.
>
> **First challenge: accuracy efficiency trade off as the zoo grows.**
> In conventional model zoo selection, more models usually mean more test time evaluation on each new task, forcing a trade off between accuracy and cost. ZooCast mitigates this by using a precompute and select scheme. Each model is summarized once into a low dimensional vector in the zoo representation library, and selection on a new task reduces to cosine similarities between task channel embeddings and this fixed set of model embeddings, followed by a light aggregation. No additional TSFM forward passes on the target task are needed at selection time, so adding models has very small marginal cost and a large zoo does not break efficiency. This design is exactly what allows ZooCast to keep its advantage in Figure 4 as the zoo grows.
>
> **Second challenge: redundant or low quality models in a growing zoo (Q3).**
> As the zoo expands, many models are inevitably redundant or outdated. ZooCast handles this with an automatic soft pruning mechanism based on model specific reliability weights `w_m`. As described around line 315 and in Appendix F.1, each model has an advantage subset, and the size and average advantage score of this subset jointly determine `w_m`. When new models are added, we update all `w_m` using the current advantage subsets. During selection, similarities between task channels and model embeddings are multiplied by `w_m`, and the final ranking comes from an error correcting aggregation over channels. Models that rarely outperform others receive small `w_m` and almost never appear in the top K, whereas strong new models such as Chronos-style or Sundial-style TSFMs quickly accumulate larger advantage subsets, gain higher weights, and are selected more frequently. Conversely, earlier and weaker VisionTS models see their weights gradually decrease and are chosen less and less often in practice. In the revised version, we will move this weighting rule from the appendix to the main text and make its role in handling redundant models explicit.
>
> ---
>
> > **W3.2: On comparisons against simple but strong alternatives**
>
> We agree that error-based meta-learners and feature-based retrieval models are natural reference points, and we would like to clarify how they relate to our setting. Most error-based meta-learners assume access to target labels (or a labeled validation set) and estimate transferability by actually training or evaluating candidate models on the target task. This conflicts with our strict zero-shot, forward-free constraint, where running many TSFMs per new task is precisely what we aim to avoid. Instead, we conceptually bracket such methods with the strong oracle-style baselines already reported in the paper (best single TSFM, All-13 ensemble, and brute-force forward-all selection), which upper bound what an error-based selector could achieve in accuracy but at much higher test-time cost.
>
> For feature-based retrieval, we already include a representative and competitive transferability baseline, namely LogME-guided ensembles, which select TSFMs using model features on the target data. On the univariate subset where LogME is applicable, ZooCast outperforms this feature-based selector while preserving zero-shot efficiency. Extending our framework with additional handcrafted time-series descriptors or more general error-based meta-learners is an interesting but orthogonal direction; our current baselines are chosen to reflect the main accuracy and efficiency trade off under the zero-shot assumptions that motivate ZooCast.
>
> ---
>
> We hope that the above clarifications adequately address your concerns about the design, training, and evaluation of ZooCast. If anything remains unclear, or if you would like to see further analyses or explanations, we would be very happy to continue the discussion.

---

> ### Author Response · Authors · 2025-11-26
> **Kindly Request for Further Feedback**
>
> Dear Reviewer or1P,
>
> I hope this message finds you well. As the author–reviewer discussion period is approaching its end, I would like to kindly check whether our responses have addressed your main concerns.
>
> If there are any remaining issues, questions, or suggestions for improvement, we would be very grateful to hear them and are more than happy to provide further clarification. Your feedback is extremely valuable to us, and we are eager to refine the paper based on your insights.
>
> If you feel that our clarifications have resolved your concerns, we would also be thankful if you could take this into account when updating your evaluation.
>
> Thank you again for your time and effort in reviewing our work.

---

### Official Review · Reviewer_etbY · 2025-11-01

**Soundness:** 3
**Presentation:** 2
**Contribution:** 3
**Rating:** 6
**Confidence:** 4

**Summary:**

This paper proposes ZooCast, a method for efficient zero-shot time series forecasting using a model zoo. It first characterizes each model's strengths via advantage subsets, which consist of data where the model excels. Then, it uses model-task co-embedding to map models and tasks into a shared space, enabling fast, accurate model selection through similarity matching.  And finally employs an error-correcting consensus mechanism that aggregates multi-channel signals into robust recommendations. The approach achieves high performance with minimal computational cost, making it scalable for real-world deployment.

**Strengths:**

1.	This paper presents a framework for zero-shot time series forecasting by introducing the novel concept of advantage subsets to characterize each model’s unique strengths. It then maps both models and tasks into a shared space, enabling fast and accurate model selection through simple similarity matching.
2.	The authors not only provide a theoretical efficiency analysis of the proposed method, but also demonstrate its practical effectiveness through runtime comparisons, confirming that their approach is indeed efficient and well aligned with the original goal of rapidly selecting optimal models for different forecasting tasks.
3.	ZooCast exhibits exceptional flexibility and scalability, enabling seamless integration of new models into the model zoo without the need to re-evaluate or recompute embeddings for existing models.

**Weaknesses:**

1.	The results in Table 2 show that ZooCast's sMAPE is higher than that of Sun.B, indicating that the ensemble performs worse than the best individual model. If the ensemble fails to outperform the top single model, what is its practical value?
2.	Since the extractor only supports fixed-length inputs (length 36), but downstream datasets are often longer than this, would the model selected from the embedding of different segments of length t sampled from the same dataset be consistent? Could sampling different segments lead to varying model rankings?
3.	The paper proposes a multi-channel voting mechanism to derive a consensus ranking. Although the authors mention accounting for noisy channel-wise similarities, further analytical experiments could be conducted to validate this design choice. Especially since, in practice, selecting the best model for each channel is a more intuitive and commonly adopted strategy. Moreover, the current approach of choosing different models per channel and then aggregating via voting may ignore intrinsic dependencies among channels, which could result in suboptimal recommendations. A deeper investigation into these aspects would strengthen the methodological justification and overall robustness of the proposed framework.

**Questions:**

1.	Please explain why ZooCast's sMAPE in Table 2 is higher than that of Sun.B, and clarify the practical value of the ensemble if it underperforms compared to the best individual model.
2.	Please analyze whether the model selected from embeddings of different segments (of length 36) sampled from the same long time series remains consistent, and discuss the potential impact of segment variation on model ranking stability.
3.	Please provide further justification for using a multi-channel voting mechanism to derive a consensus ranking, especially given that selecting the best model per channel is a more intuitive strategy. Additionally, please investigate whether the independence assumption in channel-wise model selection overlooks intrinsic channel dependencies, and assess how this might affect recommendation quality.

---

> ### Author Response · Authors · 2025-11-24
> **Discussion part 1/2:  Ensemble value and multi-channel consensus ranking**
>
> > **W1&Q1: Practical value when ZooCast’s sMAPE is slightly higher than Sun.B in Table 2**
>
> We agree that Table 2 might look puzzling at first glance: ZooCast-Top5 has a slightly higher global sMAPE than Sun.B, yet we still advocate its practical value. Our view is as follows.
>
> **(a) Rank vs. aggregated sMAPE: robustness across 97 configurations.**
> Global sMAPE is a single aggregated number and can be influenced by a few extreme datasets. Our primary robustness metric is therefore **Rank**, computed by ranking models by sMAPE within each of the 97 configurations and then averaging these ranks. Although Sun.B has the lowest aggregated sMAPE (0.430 vs. 0.431 for ZooCast-Top5), ZooCast-Top3/Top5 achieve much better average Rank (3.688 / 3.158 vs. 4.845 for Sun.B; lower is better). This means that, across diverse domains, horizons, frequencies, and variate counts, ZooCast is more consistently near the top, with the sMAPE gap to Sun.B being numerically negligible but the Rank gain being substantial. This matches our objective in Eq. (2): improving average relative performance and robustness across tasks under a limited compute budget, rather than optimizing a single global scalar.
>
> **(b) Evidence that ZooCast can outperform the best single TSFM.**
> Beyond Table 2, Figure 5 shows that ZooCast-Top3 consistently improves Rank over both All-13 and the best single TSFM across different zoo-representation sizes and advantage-subset thresholds, with only a few models in the ensemble. Figure 4 further shows that ZooCast-Top1, with the same inference cost as a single TSFM, achieves better Rank than heuristic “current best” strategies in a sequential model-release scenario. Thus, even without large ensembles, the embedding-based selection itself already brings tangible gains.
>
> **(c) Practical value despite a tiny sMAPE gap.**
> In real deployments, users do not know in advance which single TSFM (Sun.B or others) is best for a new task, and fully evaluating all models is often infeasible. ZooCast provides a **forward-free, similarity-based selector** that gets very close to the best single model in sMAPE while improving Rank and robustness across many heterogeneous tasks. In this sense, the framework remains practically valuable even if, on one aggregated benchmark metric, it is matched or very slightly outperformed by the strongest individual TSFM.
>
> ---
>
> > **W3&Q3: On multi-channel voting vs per-channel best model and channel dependencies**
>
> We appreciate this question, as it goes to the heart of how ZooCast aggregates information across channels.
>
> **(a) Why not simply pick a separate “best model” for each channel?**
> First, per-channel model selection would make **inference prohibitively expensive** in many realistic multivariate settings. A single time series can have tens or even hundreds of channels; if each channel is assigned its own TSFM, deployment would need to run many distinct models per task at test time. This defeats the purpose of ZooCast as an efficient zero-shot selector that replaces “run all TSFMs” with a few similarity lookups and a compact ensemble.
>
> Second, per-channel selection can lead to incoherent system-level behavior. In typical applications, different channels describe **different aspects of the same underlying system**, and driving each of them with an entirely different TSFM can cause inconsistencies across channels and make downstream use of the forecasts difficult. Our ECOC-style error-correcting consensus ranking is designed to aggregate a limited number of channel-level “hard” decisions into a single robust global ordering of models, trading some local flexibility for a coherent, stable model set at the task level.
>
> **(b) Does multi-channel voting assume independent channels or ignore dependencies?**
> Our ECOC-style voting does not rely on channel independence; instead, it treats each channel as a noisy view of the same task-level preference. When channels are correlated, their recommendation codes are also correlated, and the redundancy is exactly what majority voting uses to correct local noise rather than to assume independence. Moreover, channel-wise information is used only at the **selection stage**: the final forecasting is always performed by the chosen TSFM(s) on the **full multivariate input `X`**, where cross-channel dependencies are modeled directly by the TSFM architectures. In fact, assigning different TSFMs to different channels would be the approach that truly ignores multivariate dependencies, because no single model would ever see the joint signal. We will clarify this rationale in Section 4.3 and briefly compare against naive per-channel selection in the revised version.

---

> ### Author Response · Authors · 2025-11-24
> **Discussion part 2/2: Fixed-length segment sampling and ranking stability**
>
> > **W2&Q2: Consistency across sampled segments and impact on ranking stability**
>
> We appreciate the concern about using a fixed-length extractor (length 36) on long time series and the possibility that sampling different segments might lead to unstable model rankings. In ZooCast, the choice of 36 is a **deliberate compromise**: if the window were much longer, many shorter series would require extensive padding or concatenation; if it were much shorter, the segment might no longer capture representative patterns.
>
> To directly test how sensitive model selection is to segment sampling, we add two complementary experiments on a representative subset of GIFT-Eval. Because running additional analyses over all 97 configurations is computationally expensive, we select seven datasets (M_DENSE, bizitobs_l2c, ett1, jena_weather, hierarchical_sales, m4_hourly, us_births) that cover all seven domains, and keep all their configurations in frequency/variates/prediction length (20 configurations total). All deltas in the table below are measured relative to the original ZooCast setting (seed 2024, one sample per channel and task).
>
> **(a) “Seed” rows: different segments via different seeds (1 sample per task).**
> In the first block (rows marked `Seed`), we keep the sampling scheme fixed (one length-36 segment per task/channel) and vary only the random seed. The changes in Rank and sMAPE are extremely small: Top-1/Top-3 Rank shifts are at most +0.05, and sMAPE is essentially unchanged. This shows that, for a given task and channel, the co-embedding is **insensitive to which specific segment is drawn**, and model rankings are highly stable under this kind of variation.
>
> **(b) “Sample” rows: averaging multiple segments per task.**
> In the second block (rows marked `Sample`), we use seeds 2024–2028 to draw multiple segments per task, embed each, and average their embeddings to represent the task. Intuitively, this might seem more stable, but as the number of samples per task increases from 1 to 20, Top-3 Rank steadily degrades (up to +2.60) and sMAPE also worsens. This indicates that averaging over many heterogeneous segments can **wash out distinctive patterns**, causing the embedding to drift away from the true task characteristics and harming ranking quality.
>
> |     Exp.     |  Seed(s)  | Samples / task | ΔRank@Top-1 Z.C. | ΔRank@Top-3 Z.C. | ΔsMAPE@Top-1 Z.C. | ΔsMAPE@Top-3 Z.C. |
> | :----------: | :-------: | :------------: | ---------------: | ---------------: | ----------------: | ----------------: |
> |     Seed     |   2028    |       1        |            +0.00 |            -0.35 |            +0.000 |            +0.000 |
> |     Seed     |   2027    |       1        |            +0.00 |            -0.35 |            +0.000 |            +0.000 |
> |     Seed     |   2026    |       1        |            +0.05 |            +0.00 |            +0.000 |            +0.000 |
> |     Seed     |   2025    |       1        |            +0.05 |            +0.05 |            +0.000 |            +0.005 |
> | **Baseline** |   2024    |       1        |            +0.00 |            +0.00 |            +0.000 |            +0.000 |
> |    Sample    | 2024–2028 |       1        |            +0.02 |            -0.16 |            +0.000 |            +0.001 |
> |    Sample    | 2024–2028 |       2        |            +0.07 |            +0.80 |            +0.001 |            +0.011 |
> |    Sample    | 2024–2028 |       5        |            +0.21 |            +1.73 |            +0.025 |            +0.043 |
> |    Sample    | 2024–2028 |       10       |            +1.10 |            +2.39 |            +0.011 |            +0.027 |
> |    Sample    | 2024–2028 |       20       |            +0.93 |            +2.60 |            +0.010 |            +0.028 |
>
> In summary, these results show that:
>
> - With the default setting (one segment per task/channel), changing the seed (and thus the specific 36-length segment) leads to negligible changes in Rank and sMAPE, so model selection is stable across sampled segments.
> - Over-averaging many segments per task is actually harmful, as it dilutes informative patterns and degrades Rank, which empirically supports our simple, efficient sampling design.
>
> We believe this provides a direct and quantitative answer to whether fixed-length segment sampling causes severe ranking instability: in the regime used in the paper, the effect is very small, and the 36-length extractor, while not perfect, is a reasonable and practically robust design choice. At the same time, as noted in the Limitations, a fixed input length does introduce some rigidity, and we appreciate the reviewer’s suggestion; in future work, we plan to extend the extractor to support variable-length inputs while preserving the observed stability properties.

---

> ### Author Response · Authors · 2025-11-26
> **Kindly Request for Further Feedback**
>
> Dear Reviewer etbY,
>
> I hope this message finds you well. As the author–reviewer discussion period is approaching its end, I would like to kindly check whether our responses have addressed your main concerns.
>
> If there are any remaining issues, questions, or suggestions for improvement, we would be very grateful to hear them and are more than happy to provide further clarification. Your feedback is extremely valuable to us, and we are eager to refine the paper based on your insights.
>
> If you feel that our clarifications have resolved your concerns, we would also be thankful if you could take this into account when updating your evaluation.
>
> Thank you again for your time and effort in reviewing our work.

---

> ### Comment · Reviewer_etbY · 2025-11-28
>
> Thanks for the detailed response. My concerns have been addressed. I will update the evaluation accordingly.

---

### Author Response · Authors · 2025-12-01
**Summary of rebuttals**

We are very grateful to the four reviewers and the AC for their time and thoughtful discussion.

Our paper introduces **ZooCast**, to our knowledge **the first systematic model-zoo framework in time-series forecasting**: over a large TSFM zoo, it combines advantage subsets, a model–task co-embedding and an error-correcting consensus ranking to enable truly zero-shot, forward-free model selection, achieving **strong accuracy, efficiency and scalability** on GIFT-Eval.

Shared positive assessment:

- ZooCast is viewed as a meaningful model-zoo based deployment framework for TSFMs, rather than yet another single universal model (aBtv, or1P, etbY, ReTB).
- The method is regarded as technically sound with thorough experiments, including complexity analysis and systematic evaluation on GIFT-Eval (or1P, etbY, ReTB).
- Efficiency and scalability are repeatedly highlighted: model embeddings are precomputed once, new tasks only require similarity matching, and new TSFMs can be added with very low marginal cost (aBtv, ReTB, etbY).
- Substantive concerns on robustness, stability and sensitivity have all been addressed either by explicitly referring to and clarifying existing content in the main paper and appendix, or by additional ablation studies; no reviewer raised new objections after the rebuttal (aBtv, etbY, ReTB, or1P).

Per-reviewer summaries:

- aBtv (2 → 4):
  Main concerns: clarity of the overall pipeline, training of the co-embedding and the intuition behind advantage subsets.
  We clarified the prediction flow, co-embedding objectives and advantage subsets by referring to and sharpening the explanations already present in the paper, and added $\underline{\text{Ablation on noise robustness}}$ around advantage subsets; the reviewer stated “no further questions” and raised the score from 2 to 4.

- etbY (6 → 8):
  Main concerns: practical value of the ensemble versus the best single model, stability of fixed-length segment sampling and the multi-channel voting mechanism.
  We justified ensemble value in terms of Rank and robustness across 97 configurations, added an $\underline{\text{Ablation on segment-sampling stability}}$, and explained the efficiency and coherence benefits of the ECOC-style voting; the reviewer stated that all concerns were addressed and increased the score from 6 to 8.

- ReTB (6 → 6):
  Main concerns: dependence on pretraining sources for the zoo representation set `D` and the auxiliary corpus `D*`, and risks under distribution mismatch.
  We ran a series of ablation studies on the composition, scale and perturbations of `D` and `D*`, summarized as $\underline{\text{Ablation on the composition, scale and perturbations of D and D*}}$, showing only mild performance changes when diversity is preserved.

- or1P (4 → 4):
  Main concerns: the link between similarity and error, robustness under shift.
  We reinforced the existing setup with $\underline{\text{Top-K alignment analysis}}$ and $\underline{\text{Ablation on D* sensitivity}}$.

Issue types and changes to the paper:

- About half of the issues (around ten) were resolved by explicitly referring to and sharpening existing content in the main paper or appendix.
- We added five main experiments, most of which are ablations or sensitivity analyses:
  1) $\underline{\text{Ablation on segment-sampling stability}}$: effects of different seeds and numbers of segments on ranking and sMAPE;
  2) $\underline{\text{Top-K alignment analysis}}$ between similarity-based and error-based rankings: consistency between ZooCast’s ranking and the true error-induced ranking;
  3) $\underline{\text{Ablation on D* scale and perturbation sensitivity}}$ (noise, trend, seasonality);
  4) $\underline{\text{Ablation on the composition of D}}$ (different pretraining-like source combinations);
  5) $\underline{\text{Ablation on noise robustness}}$ supporting the advantage-subset design.

**All new results are consistent with and strengthen our original claims** about ZooCast’s robustness and design intuition.
- The remaining issues led to structural and presentation changes: we moved several key technical details (such as the co-embedding architecture, the transferability loss and the model-specific weights $w_m$) from the appendix into the main text, and added clearer cross-references and figures so that readers can follow the core design without heavy back-and-forth.

Rebuttal progress summary table:

| Reviewer | Initial | Final |        Discussion         | Score update         |
| -------: | :-----: | :---: | :-----------------------: | :------------------- |
|     aBtv |    2    |   4   |        🟢 completed        | 🔺 raised from 2 to 4 |
|     etbY |    6    |   8   |        🟢 completed        | 🔺 raised from 6 to 8 |
|     ReTB |    6    |   6   | 🟡 no reply after rebuttal | ➖ no explicit change |
|     or1P |    4    |   4   | 🟡 no reply after rebuttal | ➖ no explicit change |

---

### Meta-Review · Area_Chair_a6fi · 2026-01-12

**Summary:**

This paper introduces ZooCast, an interesting framework for zero-shot time-series forecasting by leveraging a model zoo. The key idea is to leverage advantage subsets and a model–task co-embedding that turns model selection into efficient similarity retrieval, with promising results on GIFT-Eval and runtime characteristics. However, key parts of the methodology are insufficiently justified, e.g., co-embedding extractor, how rankings translate into final predictions, etc., which undermines reproducibility and confidence in the key mechanism. In addition, important failure modes and robustness questions remain unresolved, e.g., cases when and where the ensemble underperforms the best single model,  and limited validation of the multi-channel consensus voting versus simpler alternatives. Overall, despite an interesting direction, this paper cannot be accepted due to unclear description of the core methodology and insufficient evidence that the proposed pipeline is robust and reliably in practice.

**Reviewer Concerns:**

The concerns over the prediction flow, co-embedding objectives and advantage subsets have been partially addressed. However, concerns are still remain about the generalization capablilty based on the advantage subsets, fixed length input, and how rankings translate into final predictions.  In addition, important failure modes and robustness questions remain not fully resolved.

**Reviewer Scores:**

One reviewer may raise score from 2 to 4. Other reviewers may maintain their scores.

---

### Decision · Program_Chairs · 2026-01-26

Reject